



# Modeling Antarctic ice shelf basal melt patterns using the one-Layer Antarctic model for Dynamical Downscaling of Ice–ocean Exchanges (LADDIE)

Erwin Lambert[1], André Jüling[1], Roderik S. W. van de Wal[2,3], and Paul R. Holland[4]

[1]Royal Netherlands Meteorological Institute (KNMI), De Bilt, The Netherlands
[2]Institute for Marine and Atmospheric Research Utrecht (IMAU), Utrecht University, Utrecht, The Netherlands
[3]Department of Physical Geography, Utrecht University, Utrecht, The Netherlands
[4]British Antarctic Survey, Cambridge, UK

**Correspondence:** Erwin Lambert (erwin.lambert@knmi.nl)

**Abstract.** A major source of uncertainty in future sea-level projections is the ocean-driven basal melt of Antarctic ice shelves. Whereas ice sheet models require a kilometer-scale resolution to realistically resolve ice shelf stability and grounding line migration, global or regional 3D ocean models are computationally too expensive to produce basal melt forcing fields at this resolution. To bridge this resolution gap, we introduce the 2D numerical model LADDIE (one-Layer Antarctic model for Dynamical Downscaling of Ice–ocean Exchanges) which allows for the computationally efficient modeling of basal melt rates. The model is flexible, and can be forced with output from coarse 3D ocean models or with vertical profiles of offshore temperature and salinity. In this study, we describe the model equations and numerics. To illustrate and validate the model performance, we apply the model to two test cases: the small Crosson-Dotson Ice Shelf in the warm Amundsen Sea region, and the large Filchner-Ronne Ice Shelf in the cold Weddell Sea. At ice-shelf wide scales, LADDIE reproduces observed patterns of basal melt and freezing that are also well reproduced by 3D ocean models. At scales of 0.5–5 km, which are unresolved by 3D ocean models and poorly constrained by observations, LADDIE produces plausible basal melt patterns. Most significantly, the simulated basal melt patterns are physically consistent with the applied ice shelf topography. These patterns are governed by the topographic steering and Coriolis deflection of meltwater flows, two processes that are poorly represented in basal melt parameterisations. The kilometer-scale melt patterns simulated by LADDIE include enhanced melt rates in basal channels, in some shear margins, and nearby grounding lines. As these regions are critical for ice shelf stability, we conclude that LADDIE can provide detailed basal melt patterns at the essential resolution that ice sheet models require. The physical consistency between the applied geometry and the simulated basal melt fields indicates that LADDIE can play a valuable role in the development of coupled ice–ocean modeling.

## 1 Introduction

Sea-level projections come with a large uncertainty, in particular those beyond 2100 (Oppenheimer et al., 2019; Fox-Kemper et al., 2021). A major component of this uncertainty is the ocean-driven basal melt of Antarctic ice shelves (Seroussi et al., 2020). Basal melt is a critical process that determines the state and fate of ice shelves. Increased basal melt can lead to a



thinning and weakening of ice shelves, which in turn reduces their buttressing effect on the ice flow of grounded ice to the ocean (e.g. Gudmundsson et al., 2019; Sun et al., 2020). Basal melt rates must thus be modeled accurately in order to simulate
ice sheet dynamics and future sea-level projections (Goldberg et al., 2019). The modeling is obstructed, however, by the high computational cost of ocean models when configured at a high resolution. As a consequence, modeled basal melt rates are typically provided at a substantially coarser resolution than that at which ice sheet model simulations are performed. In this study, we present a new basal melt model, LADDIE (one-Layer Antarctic model for Dynamical Downscaling of Ice–ocean Exchanges). This model simulates the quasi-horizontal flow of the meltwater layer beneath the ice shelf base. Using this
model, (sub-)kilometer basal melt rates can be simulated under reasonable computational cost.

Basal melt is strongly determined by the ocean dynamics in the upper ocean layer below the ice shelves. At the ice shelf base, fresh and cold meltwater is produced. Due to the salinity-dominated density contrast with surrounding seawater, this fresher meltwater rises along the slope of the ice shelf base toward the ice shelf front. The velocity of this meltwater flow, relative to the presumed stagnant ocean beneath, induces shear-driven turbulence. This turbulence maintains a net transfer of heat, salt,
and mass from the cavity waters toward the upper ocean layer. The buoyancy-driven transport and downstream modification of the meltwater is typically referred to as plume dynamics (Jenkins, 1991). Due to the turbulence-driven heat import from ambient cavity waters, the upper ocean layer is typically warmer than the local pressure-melting point. As a consequence, the plume dynamics maintain basal melt, the transport of meltwater, and a net overturning circulation. It is these dynamics of the upper ocean layer that are simulated by LADDIE.

Observations from remote sensing products have revealed clear, spatially heterogeneous patterns in basal melt rates. Compared to the ice shelf average values, basal melt rates are enhanced in specific regions. Along all Antarctic ice shelves, enhanced melt rates are observed in the deep grounding zones near the deepest parts of the grounding line (Rignot and Jacobs, 2002; Rignot et al., 2013). Detailed observations of individual grounding zones have shown that local basal melt rates can vary by an order of magnitude at horizontal distances of kilometers (Dutrieux et al., 2013; Khazendar et al., 2016; Marsh et al., 2016).
Stretching from the grounding zones to the ice-shelf front, elevated basal melt rates are also observed in narrow topographic channels in both warm and cold regions (Alley et al., 2016; Berger et al., 2017). With the exception of some very wide channels (Gourmelen et al., 2017), most have a characteristic width of a kilometer (Zeising et al., 2022). These basal channels have a preference for aligning along topographic boundaries and margins of high ice-velocity shear (Alley et al., 2019). In these shear margins, locally enhanced basal melt can initiate ice shelf damage (Lhermitte et al., 2020). Altogether, these observations have
shown that basal melt rates vary spatially at scales of kilometers across ice shelves in both warm and cold regions.

The ice mass loss contributing to sea-level rise is simulated by numerical ice sheet models. These models require basal melt rates as an external forcing (Seroussi et al., 2020). In various studies, the stability of modeled ice sheets was found to be most sensitive to basal melt in specific regions. The basal melt regions inducing the strongest ice mass loss were found to be grounding zones (Reese et al., 2018b; Goldberg et al., 2019) and shear zones (Goldberg et al., 2019; Feldmann et al., 2022).
Moreover, locally enhanced basal melt in grounding zones is identified as a necessary ingredient to reproduce the observed grounding line retreat of the Smith glacier (Lilien et al., 2019). In a state-of-the-art ice sheet model, basal melt within a kilometer-scale distance from the grounding line most strongly affected the ice sheet stability (Seroussi and Morlighem, 2018).





In a coupled ice–ocean simulation, enhanced basal melt rates in channels along the western boundaries led to a reduction in buttressing (Jordan et al., 2018). These studies suggest that the spatial pattern of basal melt is as important as average melt rates
to the ice-sheet response. In particular, an accurate basal melt forcing is required in the regions where observations indicate locally enhanced basal melt rates; these are the grounding zones, basal channels, and shear zones. The realistic simulation of these kilometer-scale basal melt patterns is therefore a key goal for ocean and ice modelers.

Basal melt rates can be simulated using 3D ocean models by explicitly resolving the circulation within ice shelf cavities (e.g., Mathiot et al., 2017). These models are well-suited to simulate the exchange of heat between the deep ocean, the continental
shelf, and the ice shelf cavities. This heat exchange is governed by, among other drivers, the Antarctic Slope Current (Thompson et al., 2018; Nakayama et al., 2021), bathymetric troughs (e.g., Wåhlin et al., 2021), and dense water formation (Morrison et al., 2020). A range of 3D models have performed well in representing the overall heat transport onto the continental shelf and into ice shelf cavities (e.g., Kimura et al., 2017; Nakayama et al., 2019; Moorman et al., 2020; Naughten et al., 2022). However, 3D ocean models are computationally too expensive to resolve kilometer-scale processes (Hewitt et al., 2022). These models
can only be feasibly configured globally at a resolution of 0.25° ($\approx$7.5 km) (e.g., Mathiot et al., 2017; Pelletier et al., 2022). At such a coarse resolution, ocean models are clearly unable to reproduce kilometer-scale features such as basal melt channels. For example, the coupled United Kingdom Earth System Model includes an ocean model at 1° resolution ($\approx$30 km near Antarctica) and an ice sheet model with 2 km resolution at the Antarctic grounding line (Smith et al., 2021; Siahaan et al., 2022). This high ice-sheet model resolution is required in particular in the grounding zone to reliably model the evolution of the grounding line
(Schoof, 2007). The example illustrates the large resolution gap between the basal melt rates that ocean models can provide, and that which ice sheet models require.

In part due to this resolution gap, numerical ice sheet models are typically forced with parameterised basal melt rates (Seroussi et al., 2020). These parameterisations are either linear or quadratic scalings of offshore temperatures and salinities (Favier et al., 2019), or built upon (quasi-)one-dimensional flow descriptions of meltwater below the ice shelf (Reese et al.,
2018a; Lazeroms et al., 2019; Pelle et al., 2019). The (quasi-)one-dimensional parameterisations may reasonably resolve enhanced melt rates in the deeper grounding zones (Favier et al., 2019; Burgard et al., 2022). Yet these parameterisations require assumptions on the horizontal flow field of meltwater plumes (Favier et al., 2019). As a consequence, the parameterised melt patterns do not account for the Coriolis deflection of this flow field (Holland and Feltham, 2006). In addition, these parameterisations do at most partly account for the topographic steering of the subshelf flow field, which is crucial to resolve the
enhanced basal melt within basal channels (Gladish et al., 2012; Millgate et al., 2013; Alley et al., 2019). Whilst being able to provide basal melt rates at a low computational cost, these parameterisations are thus ill-equipped to produce detailed spatial basal melt patterns.

In order to simulate realistic, high-resolution basal melt patterns, we here present the 2D model LADDIE (one-Layer Antarctic model for Dynamical Downscaling of Ice–ocean Exchanges). The model is built upon previous 2D (plan view) models of
the upper ocean below ice shelves (Holland and Feltham, 2006), which have been applied to the Filchner-Ronne (Holland et al., 2007), Larsen (Holland et al., 2009) and Pine Island (Payne et al., 2007) ice shelves with smooth ice shelf topographies. LADDIE is specifically designed to simulate basal melt rates that are consistent with unsmoothed ice shelf topography. The consis-





tency of the basal melt rates with the ice shelf topography makes the model suitable for studies investigating kilometer-scale ice–ocean feedbacks, such as within basal channels (Gladish et al., 2012; Sergienko, 2013; Alley et al., 2019). By resolving the 2D flow field and its Coriolis deflection and topographic steering, the model can simulate the important basal melt rates along topographic boundaries and within shear zones (Jordan et al., 2018; Feldmann et al., 2022). As the model can be configured at a sub-kilometer resolution, it can also simulate detailed basal melt rates in the important region near grounding lines (Seroussi and Morlighem, 2018). The 2D configuration prevents the model from explicitly resolving processes like turbulence and the barotropic and overturning circulations in the cavity; these processes are approximated or parameterised. The model can either be forced with 3D ocean model output, when cavities are explicitly resolved, or with vertical profiles of temperature and salinity, when insufficient cavity data is available. We present this model as a possible reduced-physics solution to bridge the resolution gap between ocean and ice sheet models. LADDIE can thus be used to force stand-alone ice sheet models with output from Earth System Models, or as a dynamical coupler between an ocean and ice sheet model.

To test the performance of LADDIE for the whole Antarctic region, we apply it to two categorically different ice shelves. The first is the Crosson-Dotson Ice Shelf, representative for warm ice shelves along West Antarctica. This ice shelf is relatively well studied through in-situ ocean observations (Jenkins et al., 2018) and remote sensing (Gourmelen et al., 2017; Khazendar et al., 2016), giving ample material for validation. The second case study is the Filchner-Ronne Ice Shelf, which is representative for cold ice shelves along most of East Antarctica. This ice shelf is also relatively well studied through in-situ observations (Nicholls et al., 2009; Hattermann et al., 2021; Bull et al., 2021), remote sensing (Moholdt et al., 2014), ocean modeling (Bull et al., 2021; Naughten et al., 2021), and tidal modeling (Padman et al., 2018; Hausmann et al., 2020).

## 2 Methods

To illustrate and validate the performance of the LADDIE model, we base this study on simulations of two ice shelves of opposite character. The first is the Crosson-Dotson Ice Shelf, a relatively small ice shelf in the warm Amundsen region. The second is the Filchner-Ronne Ice Shelf, Antarctica's second-largest ice shelf, residing in the cold Weddell Sea. In Sec. 2.1, we describe the model equations of LADDIE and their numerical implementation. In Sec. 2.2, we describe the external data used to configure, force, and validate the model. Finally, in Sec. 2.3, we describe the model setup including the geometry of the Crosson-Dotson and Filchner-Ronne Ice Shelves, the forcing applied for the simulations, and the parameter settings.

### 2.1 Model description

LADDIE is a two-dimensional model, built upon earlier 2D models of the meltwater layer beneath ice shelves (Holland and Feltham, 2006; Hewitt, 2020). The equations are based on the vertically-integrated Navier-Stokes equations, reduced by the Boussinesq approximation and a turbulence closure. In addition, the boundary conditions are partly based on parameterisations as described below. The model explicitly solves the horizontal fields of layer thickness, 2D velocity, temperature and salinity. LADDIE can be seen as the numerical 2D expansion of the 1D plume model (Jenkins, 1991; Lazeroms et al., 2019), thus resolving the Coriolis deflection and topographic steering of the flow within the meltwater layer.



### 2.1.1 Governing dynamics

At the heart of the model is a set of five equations for the conservation of the vertically integrated volume, momentum, heat and salt (see Fig. 1):

$$\frac{\partial D}{\partial t} + \nabla \cdot (D\boldsymbol{U}) = \dot{m} + \dot{e} \tag{1}$$

$$\frac{\partial DU}{\partial t} + \nabla \cdot (D\boldsymbol{U}U) - fDV = -\frac{gD^2}{2\rho_0}\frac{\partial \Delta\rho_a}{\partial x} + g_a'D\frac{\partial(z_b - D)}{\partial x} - C_d|\boldsymbol{U}|U + \nabla \cdot (A_h D\nabla U) \tag{2}$$

$$\frac{\partial DV}{\partial t} + \nabla \cdot (D\boldsymbol{U}V) + fDU = -\frac{gD^2}{2\rho_0}\frac{\partial \Delta\rho_a}{\partial y} + g_a'D\frac{\partial(z_b - D)}{\partial y} - C_d|\boldsymbol{U}|V + \nabla \cdot (A_h D\nabla V) \tag{3}$$

$$\frac{\partial DT}{\partial t} + \nabla \cdot (D\boldsymbol{U}T) = \dot{e}T_a + \dot{m}T_b - \gamma_T(T - T_b) + \nabla \cdot (K_h D\nabla T) \tag{4}$$

$$\frac{\partial DS}{\partial t} + \nabla \cdot (D\boldsymbol{U}S) = \dot{e}S_a + \nabla \cdot (K_h D\nabla S) \tag{5}$$

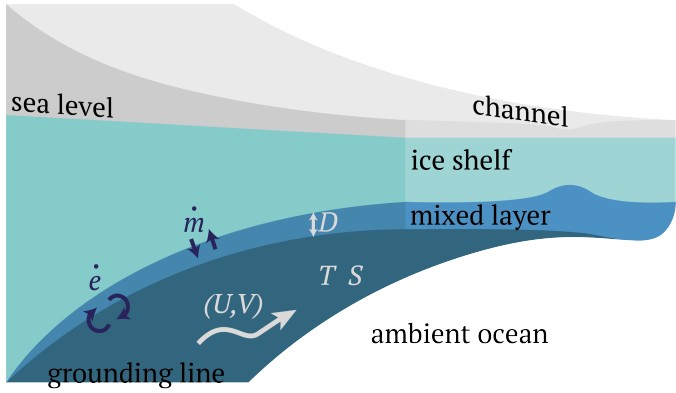

**Figure 1.** Illustration of the LADDIE model. The model solves five variables describing the meltwater layer beneath the ice shelf, denoted by the white letters. The dark blue letters denote fluxes at the top of the layer ($\dot{m}$, melt and freezing) and at the bottom of the layer ($\dot{e}$, entrainment and detrainment). The model is forced with geometric boundary conditions (grounding line and ice shelf topography) and temperature and salinity of the ambient ocean below the meltwater layer.

Here, $D$ is the layer thickness in m; $\boldsymbol{U} = (U, V)$ is the velocity vector along the ice–ocean interface, containing $U$ and $V$, the vertically averaged velocities in $x$ and $y$ directions in m s$^{-1}$; $T$ and $S$ are the vertically averaged temperature in °C and salinity in psu. Sources and sinks of layer thickness are both melting/freezing at the ice–ocean boundary, $\dot{m}$, and entrainment/detrainment at the bottom of the meltwater layer, $\dot{e}$. These are expressed as velocities perpendicular to the model's plane and their parameterisation is described in Sec. 2.1.2.

The subscript $a$ refers to variables in the motionless 'ambient' ocean below the layer, and to variables at the interface between the layer and the ambient water below. $g_a'$ is the reduced gravity:

$$g_a' = g\frac{\Delta\rho_a}{\rho_0} \tag{6}$$



$\Delta\rho_a$ is the density difference between the layer and the ambient water below the layer. We assume a linear equation of state:

$$\Delta\rho_a = \rho_0(-\alpha(T_a - T) + \beta(S_a - S)) \tag{7}$$

The ambient temperature $T_a$ and salinity $S_a$ function as forcing fields in the model. These depth-dependent variables are derived from external input, as described in Sec. 2.3.2.

The subscript $b$ refers to the ice shelf base, and variables at the ice–ocean interface. $z_b$ is the depth of the ice shelf draft in m below sea level (note: $z_b < 0$); $T_b$ is the temperature at the ice–ocean boundary.

The remaining fixed parameters are the Coriolis frequency $f$, the gravitational acceleration $g$, the reference density $\rho_0$, and the drag coefficient $C_d$. All fixed parameter values are given in Table 1. Finally, a number of parameters are used as tuning parameters or adapted to assure numerical stability under different configurations. These parameter values are listed in Table 2.

## 2.1.2 Boundary conditions

The vertical exchange of volume, heat and salt is governed by freezing or melt ($\dot{m}$ at the top boundary) and entrainment or detrainment ($\dot{e}$ at the bottom boundary). Here, we describe LADDIE's parameterisations governing $\dot{m}$ and $\dot{e}$, as well as its lateral boundary conditions.

The top boundary condition $\dot{m}$ is computed by solving the widely-adopted 'three equations' for melt and freezing comprising
the conservation of heat and salt as well as an equation of state. The latter constrains the ice–ocean boundary to remain at the freezing point. (Holland and Jenkins, 1999; Jenkins et al., 2010):

$$C_p\gamma_T(T - T_b) = \dot{m}L + \dot{m}C_I(T_b - T_i) \tag{8}$$

$$\gamma_S(S - S_b) = \dot{m}S_b \tag{9}$$

$$T_b = \lambda_1 S_b + \lambda_2 + \lambda_3 z \tag{10}$$

Here, $T_i$ is the temperature of the ice shelf interior, which functions as a forcing parameter. $C_I$ is the heat capacity of ice, $L$ is the latent heat of fusion for ice, and $T_b$ and $S_b$ are the temperature and salinity at the ice–ocean interface.

In the above parameterisation for $\dot{m}$, $\gamma_T$ and $\gamma_S$ are the turbulent exchange velocities of heat and salt in m s$^{-1}$, which we parameterise following Jenkins (1991):

$$\gamma_T = \frac{U_*}{2.12\log(U_* D/\nu_0) + 12.5 Pr^{2/3} - 8.68} \tag{11}$$

$$\gamma_S = \frac{U_*}{2.12\log(U_* D/\nu_0) + 12.5 Sc^{2/3} - 8.68} \tag{12}$$

Here, $\nu_0$ is the molecular viscosity, $Pr$ is the Prandtl number, and $Sc$ is the Schmidt number.

In the latter equations, $U_*$ is the friction velocity, defined as in Jenkins et al. (2010):

$$U_* = \sqrt{C_{d,top}(U^2 + V^2 + U_{tide}^2)} \tag{13}$$

Here, $U_{tide}$ is the mean tidal velocity in m s$^{-1}$, which functions as a forcing parameter for the model. $C_{d,top}$ is a specific
drag coefficient applied to the friction velocity used in the 'three equations' (Eq. 8). Note that, following Asay-Davis et al.





(2016), we introduce this separate drag coefficient alongside $C_d$, which is used in the momentum equations (Eq. 1). $C_{d,top}$ is commonly taken to be lower than $C_d$. This difference is imposed to correct for the presence of stratification which is neglected in the 'three equations', as discussed by (e.g., Holland and Feltham, 2006). In the current study, $C_d$ is fixed (Table 1), whereas $C_{d,top}$ is treated as a tuning parameter (Sec. 2.3.3).

The bottom boundary conditions govern the entrainment $\dot{e}$ of ambient water into the upper ocean layer. This entrainment has been described by a wide variety of different parameterisations (Jenkins, 1991; Holland and Feltham, 2006). In each of these parameterisations, entrainment is expressed as a function of the turbulent velocity in the meltwater layer. However, under conditions of weak turbulence, detrainment may occur. This process reverses the transport between the upper layer and the ambient waters, thinning the layer. Detrainment is an important process to prevent the layer thickness from growing

indefinitely. We therefore adopt the parameterisation from Gaspar (1988) describing entrainment and detrainment of dense overflows:

$$\frac{D}{2} g_b' \dot{m} + \frac{D}{2} g_a' \dot{e} = \mu U_*^3. \tag{14}$$

Here, negative values for $\dot{e}$ represent detrainment fluxes. This parameterisation is based on the balance in turbulent kinetic energy (TKE) within the meltwater layer. The source of TKE from frictional stress ($U_*$ term) is balanced by the buoyancy sink

of TKE from melting and entrainment. This parameterisation was previously used in the 3D ocean model MICOM (Holland and Jenkins, 2001) and in an earlier example of the layer model (Gladish et al., 2012). Note that the same value for $U_*$ is used, based on $C_{d,top}$, as in Eq. 8. Whether $C_d$ or $C_{d,top}$ is the more appropriate choice for Eq. 14 can be debated. However, as $\mu$ is an arbitrary constant that can be tuned accordingly, both options are equivalent.

The lateral boundary conditions in LADDIE are split between closed and open boundaries. The closed boundaries apply

at the grounding line, where the ice shelf cavity meets the grounded ice. Here, zero gradients are prescribed in $D$, $T$, and $S$. In addition, a partial slip condition is prescribed for $U$ and $V$, similar to that included in the NEMO model (Gurvan et al., 2019). This partial slip condition is a weighted average between free slip and no slip; the weighting is governed by a parameter ranging between 0 (free slip) and 2 (no slip). This parameter can be freely chosen. In the current study, we apply a value of 1, though a sensitivity analysis revealed that the parameter choice has a weak impact on the simulated basal melt patterns.

At the open boundaries, where the cavity meets the open ocean below the ice-shelf front, zero gradients are applied to all variables $D$, $T$, $S$, $U$, and $V$. These conditions allow the model to converge to a solution with a net outflow from the cavity into the deep ocean, balancing the integrated volume, heat, and salt fluxes due to entrainment and melt.

### 2.1.3   Numerical implementation

The horizontal grid is discretised in a staggered Arakawa-C grid, in which velocities $U$ and $V$ are solved at the grid cell edges,

and the other variables $D$, $T$, and $S$ are solved at grid centers. The advection of volume, heat, and salt is applied using an upstream biased advection scheme. After trial and error testing various options, this scheme was found to be most stable.

The numerical integration in time is performed using a leapfrog scheme with Robert-Asselin filter, similar to that in the NEMO model (Gurvan et al., 2019). This filter functions as a diffusion in time, described by parameter $\nu$ between 0 and 1.



**Table 1.** Model parameters. PMP = pressure melting point.

| parameter | meaning | value | unit |
|---|---|---|---|
| $f$ | Coriolis frequency | $-1.37 \times 10^{-4}$ | $\text{s}^{-1}$ |
| $g$ | gravitational acceleration | 9.81 | $\text{m s}^{-2}$ |
| $\rho_0$ | reference seawater density | 1028 | $\text{kg m}^{-3}$ |
| $C_d$ | drag coefficient | $2.5 \times 10^{-3}$ | - |
| $\alpha$ | thermal expansion | $3.733 \times 10^{-5}$ | $°\text{C}^{-1}$ |
| $\beta$ | saline contraction | $7.843 \times 10^{-4}$ | $\text{psu}^{-1}$ |
| $C_p$ | heat capacity of seawater | 3974 | $\text{J kg}^{-1} \, °\text{C}^{-1}$ |
| $L$ | latent heat of fusion | $3.34 \times 10^{5}$ | $\text{J kg}^{-1}$ |
| $C_I$ | heat capacity of ice | 2009 | $\text{J kg}^{-1} \, °\text{C}^{-1}$ |
| $\lambda_1$ | PMP salinity parameter | $-5.73 \times 10^{-2}$ | $°\text{C psu}^{-1}$ |
| $\lambda_2$ | PMP offset parameter | $8.32 \times 10^{-2}$ | $°\text{C}$ |
| $\lambda_3$ | PMP depth parameter | $7.61 \times 10^{-4}$ | $°\text{C m}^{-1}$ |
| $\nu_0$ | molecular viscosity | $1.95 \times 10^{-6}$ | $\text{m}^2\text{s}^{-1}$ |
| $Pr$ | Prandtl number | 13.8 | - |
| $Sc$ | Schmidt number | 2432 | - |
| $\mu$ | detrainment parameter | 0.5 | - |

In the NEMO model description, $\nu = 0.1$ is presented as a reference value. Using this same value in LADDIE was found to
require a relatively short time step to maintain numerical stability. We found that a value of $\nu = 0.8$ produces identical steady
state solutions and allows for a longer time step. Because in this study, we only focus on steady state solutions, we chose to
keep the value of $\nu = 0.8$. However, the choice of $\nu$ does affect the transient state, and future applications of the model should
reassess this parameter choice.

A general issue in 2D models is that the model requires a finite thickness $D$ in order to remain numerically stable. In some
regions, set by the topography of the ice shelf draft $z_b$, the velocity field tends to be divergent, particularly near grounding lines.
In some cases, the entrainment flux $\dot{e}$ is insufficient to compensate for this divergent flow, leading to a gradual thinning of the
model layer. This issue was discussed in detail by Gladish et al. (2012), who proposed the solution of imposing a minimum
layer thickness $D_{min}$. One can interpret this solution as being equivalent to the fixed upper grid thickness in 3D ocean models.
Here, we adopt this solution by increasing the local entrainment flux as required when the thickness would otherwise drop
below $D_{min}$. We treat the parameter $D_{min}$ as a tuning parameter, as described in Sec. 2.3.3.

## 2.2 External data

For the model simulations and validation, we have used a number of data sets from external sources. Here, we briefly discuss
their source, their underlying methodology, and some of their limitations.



### 2.2.1 Geometry

The realistic simulation of ice shelf basal melt rates with LADDIE strongly depends on the geometry. Here, we have taken the geometry from the Bedmachine v2 data set (Morlighem et al., 2020). This data set combines remote sensing products of the Antarctic ice surface elevation and velocity into a consistent map of ice thickness and bed topography below grounded ice. For the floating ice shelves, the ice shelf thickness is based on the assumption of hydrostatic equilibrium, a modeled firn layer, and an additional firn-depth correction. Within 3 kilometers from the grounding line, smoothing of the ice shelf thickness is applied

to ensure continuity.

Whilst providing a detailed, high resolution geometry, the Bedmachine data set contains some limitations which are relevant to our study. The smoothing within 3 km from the grounding line removes detailed topographic features in this region. In addition, the grounding line is static, which prevents the modeling of basal melt nearby retreating grounding lines. At small scales, the assumption of hydrostatic equilibrium may also be violated by a significant internal stress, obscuring the topography

of basal channels. Finally, the data set provides a snapshot of the highly dynamic ice shelf front and shear margins. This allows LADDIE to simulate instantaneous basal melt rates, but prevents it from simulating time-mean basal melt rates over a multi-year period.

The above limitations hamper the direct validation of modeled basal melt rates to observations. As the aim for LADDIE is to simulate basal melt patterns consistent with the provided topography, we will not purely validate LADDIE to observations. In

addition, we will qualitatively infer reasonable basal melt patterns for the given topography. For example, the presence of a basal channel in the applied geometry should impact the simulated flow – and possibly the basal melt rate – by LADDIE, regardless of whether this channel is a realistic feature or an artefact of the topography. In other words, an unrealistic topographic feature should induce an unrealistic yet physically consistent basal melt pattern by LADDIE.

### 2.2.2 Observations

To quantitatively validate LADDIE, we compare the simulated basal melt patterns to basal melt rates derived from remote sensing products. For simplicity, we refer to these basal melt rates as 'observations', stressing that they are highly uncertain and based on a range of assumptions. Therefore, these observations should not be interpreted as a ground truth.

For the Crosson-Dotson Ice Shelf, we validate LADDIE to observations from Gourmelen et al. (2017); Goldberg et al. (2019). These basal melt rates are derived from CryoSat-2 radar altimetry over the period 2010-2016. Changes in surface

elevation are processed using a Lagrangian analysis based on observed surface ice velocities. These changes are then combined with modeled surface mass balance and the assumption of hydrostatic equilibrium to provide basal melt rates.

Besides uncertainties in the underlying observations, the assumption of hydrostatic equilibrium induces a large uncertainty near grounding lines. This uncertainty was assessed by Milillo et al. (2022) for the Crosson Dotson Ice Shelf and was found to be small in the Kohler grounding zone, yet large in the Smith grounding zone. Another issue with observations in the

grounding zone is the timing and the relatively long period over which basal melt rates are assessed. The Kohler, Smith, and Pope grounding lines are known to have retreated at a high rate over the last decade (Scheuchl et al., 2016; Milillo et al., 2022).





This makes average basal melt rates over a multi-year period in this region less representative of instantaneous basal melt rates as simulated by LADDIE.

For the Filchner-Ronne Ice Shelf, we use the Antarctic-wide observations from Adusumilli et al. (2020). The underlying
methodology is similar to that of Gourmelen et al. (2017); it is based on a combination of CryoSat-2 altimetry, satellite-derived surface ice velocities, and modeled firn layer. The firn density model used in this study is of a coarse resolution (12.5 km) compared to the grid size of the final product (500 m). In addition, no recalibration based on local observations is applied as in Gourmelen et al. (2017). Together, these differences lead to a poorer spatial detail in the 'observed' basal melt fields. However, comparison to other remote sensing estimates (Rignot et al., 2013; Moholdt et al., 2014) gives confidence in the
assessed large-scale pattern of basal melting and freezing.

### 2.2.3 3D model reference

In this study, we introduce LADDIE as a new model to simulate high-resolution spatial basal melt patterns. To test its added value compared to 3D numerical ocean models, we compare the simulated basal melt fields for Crosson-Dotson and Filchner-Ronne to regional 3D model output.

For the Crosson-Dotson Ice Shelf, we compare basal melt from LADDIE to that simulated by a regional setup of MITgcm of the Amundsen region with resolved ice shelf cavities (Naughten et al., 2022). The ocean/sea-ice model is configured with a horizontal grid size ranging from 5.2 km offshore to 2.8 km at its shouthernmost point. The geometry is based on the BedMachine v2 geometry and the model is forced with ERA5 atmospheric reanalysis (Hersbach et al., 2020). The model was extensively tuned to reproduce observations in the region. In particular, the ocean conditions and their interannual variability in
front of Dotson and Pine Island Ice Shelves were tuned to match observations by varying sea-ice parameters, and then the melt rates for these ice shelves were tuned to match observations by selecting melting parameter values that best fit the entire region (Naughten et al., 2022). For an optimal comparison to the observations from Rignot et al. (2013), we extract and average the period 2003–2008 from the MITgcm simulation.

For the Filchner-Ronne Ice Shelf, we compare the basal melt rates from LADDIE to regional simulations using NEMO
(Hausmann et al., 2020). NEMO, similar to MITgcm, is an advanced ocean/sea-ice model, with explicitly resolved cavities. The regional simulation of the Weddell Sea was configured at a horizontal resolution varying between 4.5 km offshore to 1.5 km at the southernmost point. The model was forced with atmospheric climatology from CORE-2 (Large and Yeager, 2009), and tidal conditions at the open boundaries from FES2012 (Carrere et al., 2013). The model was specifically designed to simulate tides in the Filchner-Ronne cavity and their impact on basal melt and freezing.
Both 3D ocean/sea-ice models are configured using a fixed vertical grid. The vertical grid cell thickness below the ice shelves ranges between 10 and 50 m for MITgcm and between 10 and 150 m for NEMO. In both models, basal melt and freezing is parameterised using a similar 'three-equation' formulation as used in LADDIE (Eq. 8).

### 2.3 Model setup

In this section, we describe the model setup for the simulations of the Crosson-Dotson and Filchner-Ronne Ice shelves (Fig. 2).

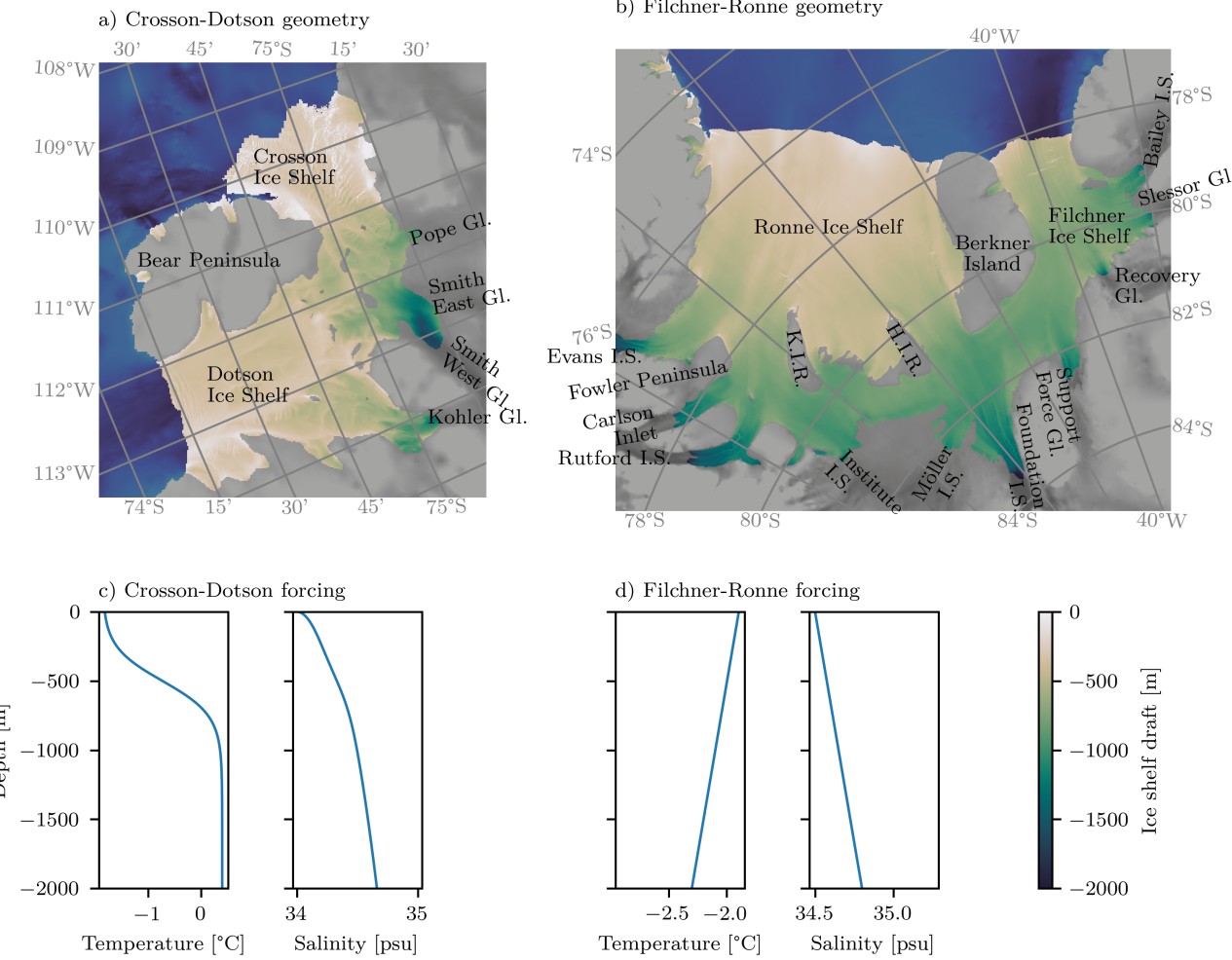

**Figure 2.** Geometry and forcing for the Crosson-Dotson and Filchner-Ronne Ice Shelves. a,b) Ice shelf geometries. The bright shading indicates the ice shelf draft, increasing from the grounding line to the ice-shelf front. The low-contrast shading denotes the bed topography below the open ocean (dark blue) and the grounded ice sheet (grey), with dark colours indicating a deep bed. Abbreviations are: Gl. (Glacier), I.S. (Ice Stream), K.I.R. (Korff Ice Rise), and H.I.R. (Henry Ice Rise). c,d) Idealised forcing profiles applied in the main simulations.

### 2.3.1 Geometry


The geometry of the model domain is given by the topography of the ice shelf draft $z_b$ and the classification of the lateral boundaries. We base our geometry on the BedMachine v2 data set (Morlighem et al., 2020), which provides an ice shelf draft





topography at a 500×500 m$^2$ grid for the entire Antarctic ice sheet. The classification of the lateral boundaries is based on the embedded mask of grounded ice, floating ice, and open ocean; from this mask, lateral boundaries are separated into open

boundaries (the ice-shelf front) and closed boundaries (the grounding line).

Due to its small size, the Crosson-Dotson Ice Shelf can be configured on the original resolution of 500×500 m$^2$. Note that previous studies have smoothed the ice shelf topography, in part to ensure numerical stability (e.g., Holland et al., 2007), which is a common practice in ice–ocean modeling (Asay-Davis et al., 2016). In order to resolve kilometer-scale basal melt features, LADDIE is configured such that it can deal with unmodified ice shelf topography.

For the Filchner-Ronne Ice Shelf, simulations at 500×500 m$^2$ are computationally demanding, as the LADDIE model is not yet coded in parallel. Hence, this ice shelf is configured at a 1000×1000 m$^2$ resolution by averaging the topography of 2×2 neighbouring grid cells.

### 2.3.2 Forcing

The LADDIE model is forced by four variables: the ambient temperature and salinity, $T_a$ and $S_a$; the ice shelf temperature $T_i$;

and the tidal velocity $U_{tide}$. In this study, we choose to focus mainly on idealised forcing fields. This choice allows for easier interpretation of the results and for a cleaner assessment of the impact of geometry on the resultant basal melt fields. To test the sensitivity of the basal melt patterns to the applied forcing, we include a comparison for Crosson-Dotson to a more realistic forcing (App. A).

For both Crosson-Dotson and Filchner-Ronne Ice Shelves, we apply uniform fields of $T_i$ and $U_{tide}$. The values are listed in

Table 2. We note that $U_{tide}$ has a large impact on the heat transfer toward the ice shelf base. In particular for Filchner-Ronne, significant improvements may be possible when changing $U_{tide}$ to a spatially variable field (Padman et al., 2018; Hausmann et al., 2020).

For $T_a$ and $S_a$, the main results in this paper are based on idealised vertical profiles of temperature and salinity. This is done for two reasons. First, as described above, an idealised forcing eases the interpretation of the impact of geometry on

the simulated basal melt patterns. Second, 1D vertical profiles resemble the output of Earth System Models without resolved cavities; we anticipate that forcing with such ESM output is a main application of the model. Below, we describe the prescribed temperature and salinity fields used as forcing. From these steady fields, at each model time step, temperatures and salinities at each grid cell are extracted at the depth of the layer base, $z_b - D$. These temperatures and salinities form the horizontal 2D fields of $T_a$ and $S_a$ at the layer base.

For the main simulations of the Crosson-Dotson Ice Shelf, we create vertical 1D profiles mimicking the observed profiles in front of the Dotson Ice Shelf (Jenkins et al., 2018). Temperature is described by a tangent hyperbolic function, ranging from -1.7°C at the surface to +0.4°C at 1200 m depth (Fig. 2c). The thermocline is centered at 500 m depth with a scale factor of 250 m. This temperature profile describes the observed two-layer stratification in front of the ice shelf. Salinity increases from 34 psu at the surface to 34.5 psu at 1200 m depth. The profile is derived from a prescribed density profile, which is quadratic

with depth; this density profile mimics the observed strongest stratification near the surface.





To test the sensitivity of the simulated Crosson-Dotson basal melt patterns to the applied forcing, we also include a simulation with 3D forcing (see App. A). As it is considered the most realistic option, this 3D forcing is also used for the model tuning (see App. B). The 3D fields of temperature and salinity are derived from the simulations with MITgcm (see Sec. 2.2.3) averaged over the period 2003–2008. This period aligns with the observation period of Rignot et al. (2013), which is used to tune LADDIE.

From the 3D model output within the cavity, the upper two grid cells below the ice shelf are removed, as these represent the meltwater layer rather than the ambient waters. To extract ambient temperatures and salinities throughout the cavity, the remaining 3D fields are regridded onto the LADDIE grid, extrapolated vertically to the ice shelf base, and horizontally into missing grid cells. Finally, to ensure smooth and continuous profiles of temperature and salinity, a vertical smoothing is applied with a window of 10 meters.

For the Filchner-Ronne Ice Shelf, we apply idealised profiles of temperature and salinity approximating the average conditions within this cold cavity. These profiles are similar to those applied to the Filchner-Ronne simulations by Holland et al. (2007). Temperature decreases linearly from freezing point at the surface to -2.3°C at 2000 m depth. Salinity increases linearly from 34.5 psu at the surface to 34.8 psu at 2000 m depth (Fig. 2d).

### 2.3.3 Parameter settings

With the geometry and forcing described above, the model setup is completed by a number of parameter choices. A number of parameter values remain unspecified in order to adapt them to different simulations. These are the time step $\Delta t$, the horizontal viscosity $A_h$ and the horizontal diffusivity $K_h$; each of these affects the numerical stability of the model and depend on the chosen resolution. Finally, two parameters are considered tuning parameters: $C_{d,top}$ and $D_{min}$.

Based on trial and error, we have determined values of $A_h$ of 100, 50, and 25 m$^2$ s$^{-1}$ to be suitable for grid resolutions
of 2000, 1000, and 500 m, respectively. Further sensitivity runs indicated that the choice of $K_h$ has a small effect on the resultant basal melt rates; for convenience, we take $K_h$ equal to $A_h$. Finally, values for the time step $\Delta t$ are determined per configuration in order to maximise the time step while ensuring numerical stability. The values for the main simulations of the Crosson-Dotson and Filchner-Ronne Ice Shelves are shown in Table 2.

We have chosen the remaining two parameters as tuning parameters, namely the drag coefficient used in the equation for the
friction velocity $C_{d,top}$ and the minimum layer thickness $D_{min}$. $C_{d,top}$ is commonly used for tuning the basal melt parameterisation in ocean models (Holland and Feltham, 2006); values in the literature vary between $1.0 \times 10^{-3}$ (Jourdain et al., 2017; Mathiot et al., 2017) and $9.7 \times 10^{-3}$ (Jenkins et al., 2010). $D_{min}$ is treated as a tuning parameter, as its impact on basal melt rates is found to be relatively large. Here, we have considered values between 2 and 8 meters.

The model is tuned to a number of observational based indicators for the Crosson-Dotson Ice Shelf. The resultant tuning
parameters are applied to the Filchner-Ronne Ice Shelf in order to test whether the tuning is valid in both cold and warm environments. Five diagnostics are derived from simulations with 3D forcing: ice-shelf average melt, melt in the Kohler grounding zone, melt along the wide Dotson Channel and along its center line, and the total overturning circulation. These diagnostics are compared to a number of remote sensing and in-situ observational sources. For horizontal grid sizes of 500×500, 1000×1000 and 2000×2000 m$^2$, values of $D_{min}$ and $C_{d,top}$ are determined such that each of the five diagnostics fall within the assessed





uncertainty ranges. The complete tuning procedure is described in Appendix B, and the values used in our simulations are shown in Table 2.

**Table 2.** Simulation-specific parameter values applied the Crosson-Dotson and the Filchner-Ronne Ice Shelves.

| parameter | meaning | Crosson-Dotson | Filchner-Ronne | units |
|---|---|---|---|---|
| $\Delta x$ | resolution | 500 | 1000 | m |
| $T_i$ | ice temperature | -25 | -25 | °C |
| $U_{tide}$ | tidal velocity | 0.01 | 0.1 | m s$^{-1}$ |
| $A_h$ | hor. viscosity | 25 | 50 | m$^2$ s$^{-1}$ |
| $K_h$ | hor. diffusivity | 25 | 50 | m$^2$ s$^{-1}$ |
| $C_{d,top}$ | exchange coefficient | $1.1\times10^{-3}$ | $1.1\times10^{-3}$ | |
| $D_{min}$ | minimum thickness | 2.8 | 4.4 | m |
| $\Delta t$ | time step | 120 | 240 | s |
| | equilibration time | 30 | 720 | model days |
| | computation time | 0.5 | 60 | CPU hours |

### 2.3.4 Experimental design

Using the configuration and parameter settings above, the equations are integrated forward in time. The layer is initialised at rest, $U = V = 0$, and the initial layer is uniformly 10 m thick. For temperature and salinity, $T$ and $S$, initial values are taken

0.1°C and 0.1 psu below the ambient values $T_a$ and $S_a$ respectively. This small initial difference in salinity ensures a stable stratification.

The current model setup is designed to simulate basal melt rates under steady state conditions, with a fixed geometry and fixed ambient conditions. The results presented in this study are thus steady state basal melt rates. For the Crosson-Dotson Ice Shelf, basal melt rates are averaged over the last 5 model days; for Filchner-Ronne over the last 10 model days. The

equilibration time, as well as the computation time required for a full spin up, is listed in Table 2.

## 3 Results

In this section, we present the steady-state basal melt patterns simulated by LADDIE for the Crosson-Dotson and the Filchner-Ronne Ice Shelves consecutively. For each ice shelf, we begin with discussing the large-scale ice shelf-wide pattern of melt and freezing. We then continue to discuss detailed spatial melt patterns in specific regions of both ice shelves. The simulated melt

patterns are compared to observations and 3D model output. For each melt feature, we assess the reliability of observations, the accuracy of LADDIE and the 3D model, and the implications for the stability of the ice shelves.



### 3.1 Crosson-Dotson

### 3.1.1 Large-scale patterns

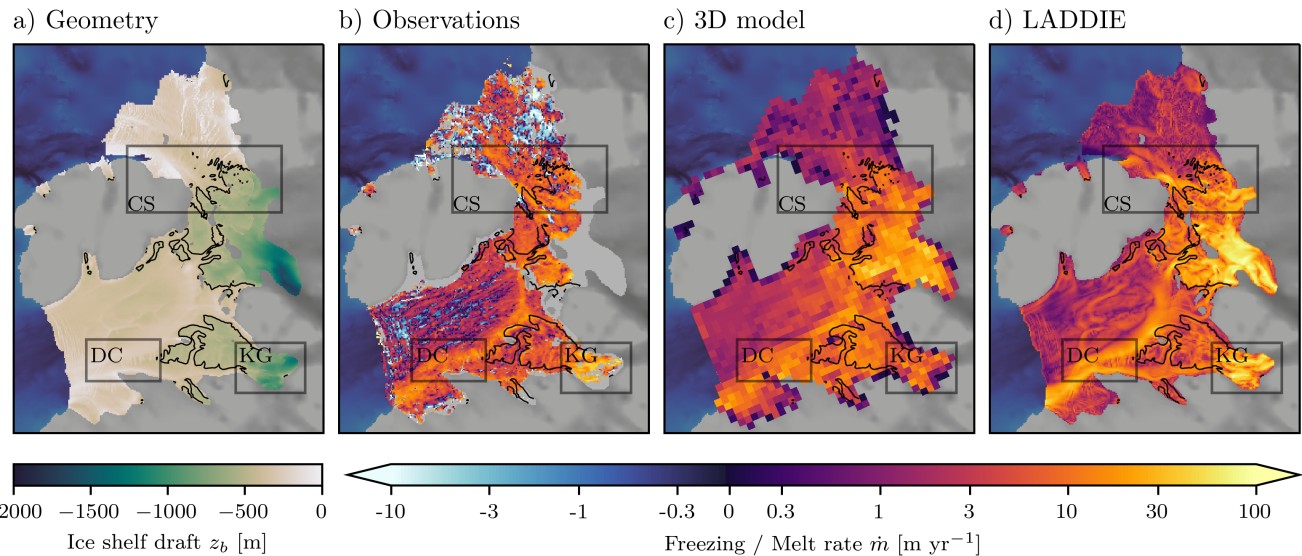

**Figure 3.** Spatial basal melt field of the Crosson-Dotson Ice Shelf. a) Geometry in terms of ice shelf draft $z_b$. b) Observed basal melt rates from altimetry over the period 2010–2016 (Gourmelen et al., 2017; Goldberg et al., 2020). c) Basal melt rates from the 3D model MITgcm. d) Simulated basal melt rates using LADDIE. The black contour line marks the ice shelf draft with a depth of 450 m. The colourscale is a symmetrical log-scale, and is linear between values -0.3 and 0.3 m yr$^{-1}$. The insets refer to KG (Kohler Grounding zone), DC (Dotson Channel), and CS (Crosson Shear margin); these regions are displayed in more detail in subsequent figures.

Across the Crosson-Dotson Ice Shelf, the highest melt rates are found in the regions where the ice shelf extends below approximately 450 meters (Fig. 3b). This corresponds with the approximate depth of the thermocline, separating deep warm water from cold shallow water (Jenkins et al., 2018). These highest melt rates can thus, to first order, be explained by the local thermal forcing, which is the temperature difference between cavity waters and the freezing temperature at the ice shelf base (e.g. Favier et al., 2019).

In the regions of the ice shelf shallower than 450 meters, a weak thermal forcing induces low basal melt rates (Fig. 3b). Even though some shallow regions appear to show basal freezing, this is an artefact of the calculation procedure and is considered unrealistic by Goldberg et al. (2019). Overall, the contrast between high melt in the deeper regions and low melt in the shallow regions is well-simulated by both the 3D model MITgcm (Fig. 3c) and by LADDIE (Fig. 3d).

This large-scale pattern is violated by one major feature, the Dotson Channel (DC). This channel is approximately 5 km wide, and extends from the deep parts of the ice shelf to the ice-shelf front (Gourmelen et al., 2017). Such channels arise due to meltwater plumes which advect relatively warm water from the deep ice shelf regions to the shallow regions (Payne et al.,





2007). The accurate modeling of such melt channels therefore requires the explicit simulation of the flow below the ice shelf. As a consequence, these melt channels cannot easily be simulated using models of fewer than 2 dimensions. Both MITgcm and LADDIE reproduce this feature (Fig. 3c,d).

### 3.1.2 Kohler grounding zone

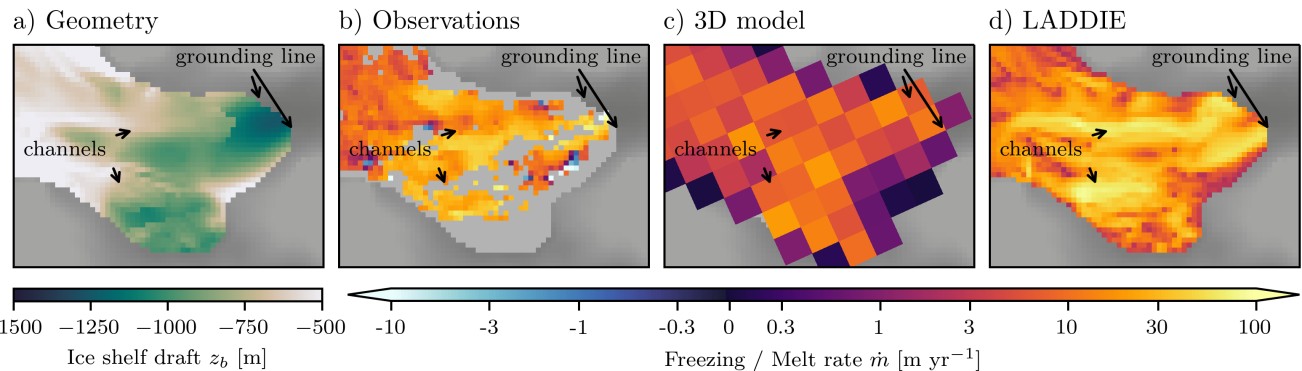

**Figure 4.** Kohler grounding zone melt. Detailed features within the inset KG in Fig. 3. Note the different colourscale for the ice shelf draft to highlight topographic details.

To discuss the more detailed melt patterns, we zoom in on several key regions. The first of these is the Kohler grounding zone (Fig. 4). The average melt rates derived from remote sensing are approximately 25 m yr$^{-1}$ (Fig. 4b, Gourmelen et al. (2017)). These rates agree well with the average melt rates of 20 m yr$^{-1}$ from airborne surveys over the same period (Khazendar et al., 2016). These melt rates are approximately 3–4 times higher than the average melt rates over the whole Crosson-Dotson Ice Shelf over the same period (Gourmelen et al., 2017). As this ratio was one of the tuning targets for LADDIE (App. B), the av-
erage melt rates in the Kohler grounding zone are well simulated by this model. In contrast, the MITgcm model underestimates the basal melt rates in this region, simulating comparable melt rates to the Crosson-Dotson average values.

     Remote sensing products do not provide reliable estimates for basal melt at the grounding line, as the assumption of flotation fails at this point. However, numerical ice sheet modeling has been used to conclude that high basal melt rates at the grounding line are a necessary ingredient to reproduce the observed grounding line retreat of the Kohler and Smith glaciers (Lilien et al.,
2019). The grounding line melt rates simulated by MITgcm are near-zero due to the lack of a simulated barotropic flow through the thin water column. In contrast, high melt rates are simulated by LADDIE at the deepest points of the grounding line, as this model is unrestricted by the water column thickness. The aforementioned modeling study suggests that significant, non-zero basal melt rates may be present at the deepest grounding lines. However, observation-based evidence is required to decide which basal melt pattern (MITgcm or LADDIE) is most realistic in this region.





Through the middle of the grounding zone, LADDIE simulates several elongated melt features. This agrees with the high spatial variability in basal melt rates observed in both the Kohler and Smith grounding zones (Khazendar et al., 2016). These enhanced melt rates align with topographic channels (Fig. 4a). The melt channels appear not to be connected to the grounding line itself, but originate downstream from the grounding line. Although this may well be an artefact of the smoothing applied to the topography (Sec. 2.3.1), it agrees with the assessment by Alley et al. (2016) that melt channels in this warm region are

predominantly ocean-sourced. Such ocean-sourced channels require locally enhanced basal melt to originate and persist. We see that LADDIE simulates enhanced basal melt in these channels, indicating that the modeled basal melt patterns can support ocean-sourced basal channels.

### 3.1.3   Dotson Channel

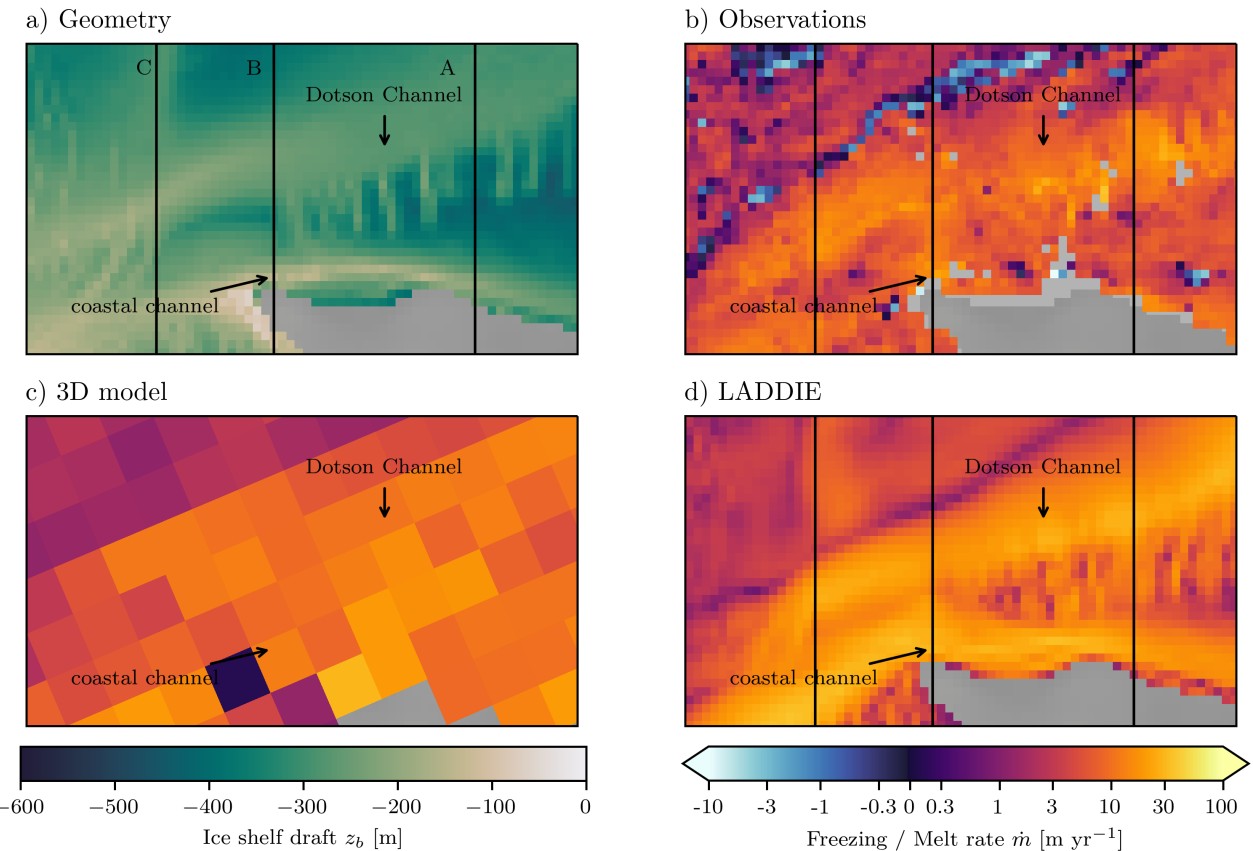

**Figure 5.** Dotson Channel melt rates. Detailed features within the inset DC in Fig. 3. Note the different colourscale for the ice shelf draft.

Next, we zoom in on the Dotson Channel and its surroundings (Fig. 5). Comparing the centerline of the topographic channel

(Fig. 5a) to the centerline of the observed melt plume (Fig. 5b) reveals a misalignment. The highest melt rates are observed on the western side of the topographic channel. Along the eastern flank of the channel, melt rates are near-zero. This western



intensification hints at a significant impact of the Coriolis deflection. The location of the melt peak with respect to the topographic channel is highlighted in Fig. 6 for three cross-sections. In each of the three sections, the Dotson Channel melt peak from both observations and LADDIE are similarly intensified on the western flank.

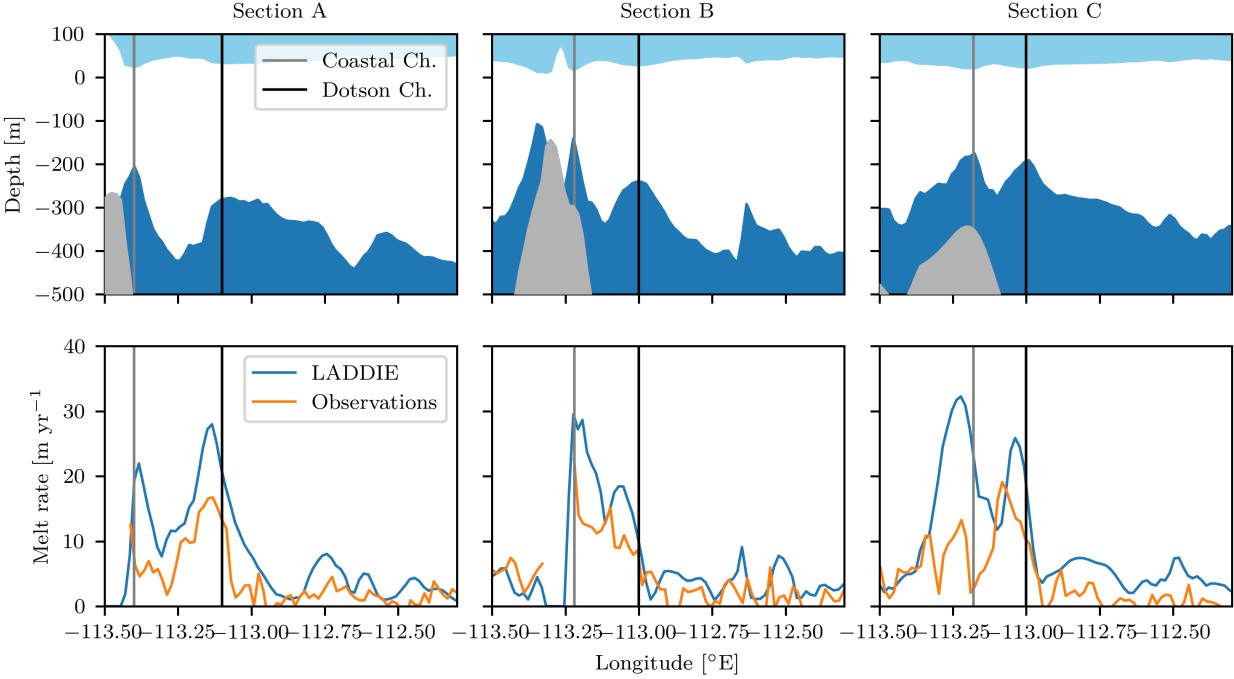

**Figure 6.** Cross sections across the Dotson and Coastal Channels. The location of sections A, B, and C are indicated in Fig. 5. Top row: geometry including the ice shelf (white), ocean cavity (blue), and solid earth (grey). The vertical lines indicate the topographic locations of the channel centrelines. Bottom row: Observed and simulated basal melt rates across the same sections.

Due to ice–ocean interactions, this western intensification can lead to a westward migration of the channel (Gladish et al., 2012). The asymmetric shape of the topographic channels, with a steeper slope on the left than on the right, indicates that the western intensification of basal melt has already impacted the ice shelf basal topography (Sergienko, 2013). As the location of the melt peak simulated by LADDIE agrees well with that of the observed melt peak, we are confident that the 2D flow of meltwater below the ice shelf, and its Coriolis deflection, is realistically simulated by LADDIE. Due to the relatively coarse resolution of MITgcm, this westward intensification cannot be simulated by this 3D model.

Closer to the coast of the Dotson Channel, the ice shelf topography reveals another channel, which we will call the Coastal Channel (Fig. 5a). This channel appears to form a boundary current that converges with the main Dotson Channel flow near the ice-shelf front. Along this coastline, observations are again uncertain, as they rely on the assumption of flotation. It is therefore difficult to detect a clear melt signal along this coastal channel (Fig. 5b). In addition, the geometry is also uncertain as it is



within 3 km from the grounding line. However, the cross-sections reveal a clear topographic Coastal Channel (Fig. 6). Across sections A and B, this channel is sharp and the simulated melt peak by LADDIE aligns with the centerline of the topographic channel, as the coastal boundary prevents a westward intensification. Across section C, the Coastal Channel has detached from the coast and has widened. Across this section, both observations and simulations show a similar westward intensification of the Coastal Channel melt peak, comparable to the Dotson Channel. Whether the Coastal Channel is realistic or an artefact of uncertainty in the topography cannot be determined in this study. However, the sharpness of the channel topography and of the melt peak may indicate that ice-ocean interactions have played a role in shaping the channel topography; if this is the case, LADDIE simulates a physically consistent narrow melt channel. As theses elevated melt rates align with the shear margin of the Dotson Ice Shelf, they have a large potential to affect the ice shelf stability.

The Dotson and Coastal Channels function as conduits for meltwater produced in the deep grounding zones. Looking back at the ice shelf wide melt patterns by LADDIE (Fig. 3d), we can distinguish two pathways. The Dotson Channel appears to provide a pathway for meltwater from the Smith grounding zone. Similarly, the Coastal Channel appears to act as the main pathway for meltwater from the Kohler grounding zone. Near the ice-shelf front, these meltwater plumes converge before exiting the ice shelf cavity. Although these pathways are clearly carved in the ice shelf topography (Fig. 3a), the remote sensing estimates of basal melt (Fig. 3b) do not provide conclusive evidence for their existence.

### 3.1.4 Crosson shear margin

The last melt pattern we highlight on the Crosson-Dotson Ice Shelf is the Crosson shear margin (Fig. 7). Ice shelf shear margins are often strongly damaged. As a consequence, the assumption of continuity – central to the translation of altimetry to basal melt rates – breaks down. The observations in shear margins are therefore unreliable. Yet indirect evidence from polynya activity points to the presence of a melt channel along the western boundary of the Crosson ice shelf, extending to the ice-shelf front (Alley et al., 2016). Indeed, away from the ice-shelf front, observations reveal enhanced melt rates in the shear margin along the coastline of Bear Peninsula (Fig. 7b).

This enhanced melt along the Bear Peninsula is simulated by both MITgcm and LADDIE (Fig. 7c,d). Tracing this melt peak simulated by LADDIE upstream toward the grounding line, we find its origin in the deep Pope grounding zone (Fig. 7d). A meandering melt channel provides a pathway for a melt plume from the Pope grounding line, past the coastline of Bear Peninsula, all the way to the ice-shelf front. Near the ice-shelf front, enhanced melt rates are observed that hint at a warm melt plume that has the potential to form a polynya. In contrast, the MITgcm model simulates no melt at the ice-shelf front and the melt channel is directed away from the shear margin toward the interior of the ice shelf (Fig. 7c).

The enhanced melt rates in shear margins can lead to thinning and weakening of the ice shelf; this results in an increase in damage and a reduction in the buttressing effect. In addition, enhanced melt rates at the ice-shelf front, as indirectly observed by polynya activity, can facilitate a retreat of the ice-shelf front. The relatively far inland position of the Crosson ice-shelf front near Bear Island (Fig. 3a), compared to the rest of the Crosson ice front, hints at high ice-shelf front retreat rates in the past. An accurate simulation of the melt plume pathway from grounding line to ice-shelf front through shear margins is therefore essential for the realistic assessment of ice-shelf stability.





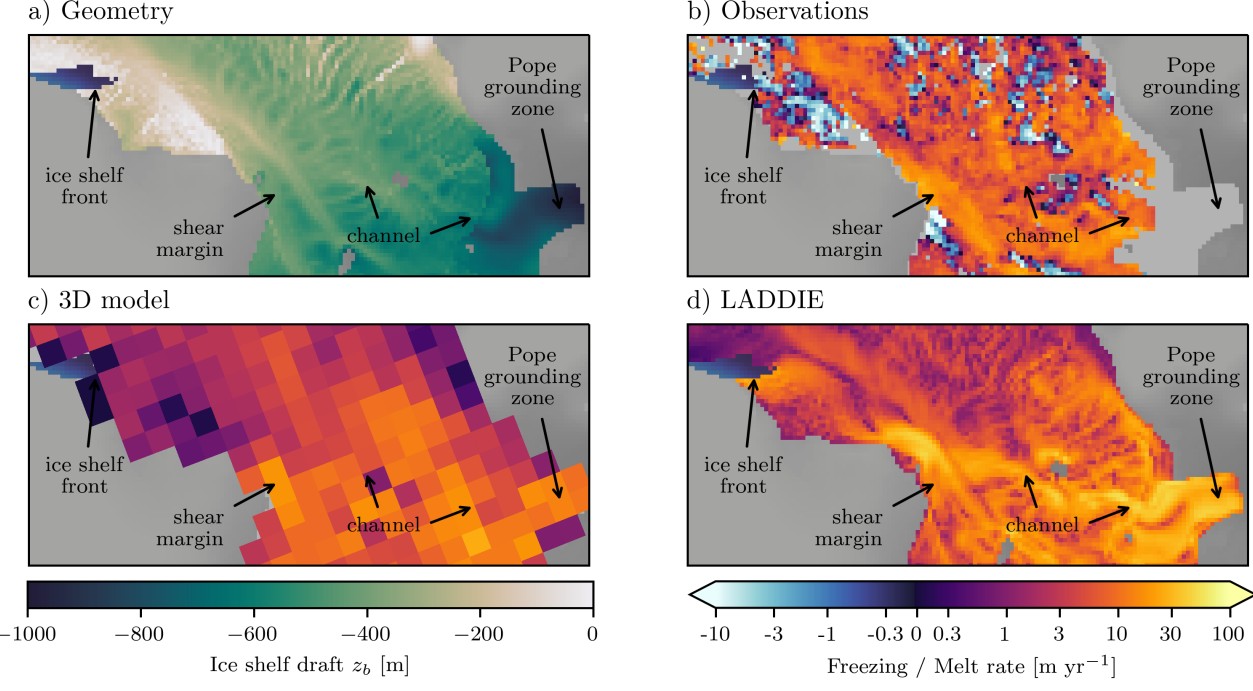

**Figure 7.** Crosson Shear margin melt. Detailed features within the inset CS in Fig. 3. Note the different colourscale for the ice shelf draft.

The pathway of meltwater from the Pope grounding zone illustrates a separation between the Crosson and Dotson cavities.

As discussed in Sec. 3.1.3, the meltwater from the Smith grounding zone exits the cavity through the Dotson Channel. However, the ice flow from both Pope and Smith glaciers exit through the Crosson ice shelf. This separation creates an indirect connection between the Pope and Smith glaciers. The meltwater plume from the Pope glacier leads to a weakening of the Crosson shear margin, reducing buttressing and enhancing the ice flow from both Pope and Smith glaciers. The meltwater pathways thus introduce an inter-dependency of the stability of these two glaciers.

## 3.2 Filchner-Ronne

### 3.2.1 Large-scale patterns

The basal melt rates of the Filchner-Ronne Ice Shelf are considerably lower than those of the Crosson-Dotson Ice Shelf. Observations indicate a mean melt rate of $0.32\pm0.1$ m yr$^{-1}$ (Rignot et al., 2013). This mean melt rate is well-reproduced by both the 3D model NEMO ($0.33$ m yr$^{-1}$, Hausmann et al. (2020)) and by LADDIE ($0.26$ m yr$^{-1}$).

Similar to Crosson-Dotson, the highest basal melt rates are found in the deep grounding zones (Fig. 8b). In contrast to Crosson-Dotson, however, these grounding zone melt rates are not driven by the presence of warm Circumpolar Deep Water.







**Figure 8.** Spatial basal melt field of the Filchner-Ronne Ice Shelf. a) Geometry in terms of ice shelf draft. b) Observed basal melt rates from altimetry Adusumilli et al. (2020). c) Basal melt rates from the 3D model NEMO including tides Hausmann et al. (2020). d) Simulated basal melt rates using LADDIE. WB: Western Boundary; RG: Rutford Grounding zone.

The Filchner-Ronne Ice Shelf cavity is isolated from intrusions of this warm water mass and is instead forced by waters at the surface freezing point, formed by sea-ice formation offshore (Nicholls et al., 2009); it is therefore classified as a cold cavity. The high melt rates in the grounding zone are caused by the low freezing point at depth. This freezing point, which can reach values of -2.5°C, is substantially lower than the surface freezing point of approximately -1.9°C. The ambient waters within the Filchner-Ronne cavity consist of a mixture of water at surface freezing temperature, and melt water created at depth. This mixture is warmer than the local freezing temperature within the deep grounding zones. This temperature difference creates a thermal forcing leading to melt rates within the grounding zone of rates up to 10 m yr$^{-1}$ (Rignot and Jacobs, 2002).



As the meltwater at these coldest temperatures rises toward the ice-shelf front, the local freezing point increases. At a certain
depth, the meltwater, mixed with ambient water due to entrainment, is colder than the local freezing freezing point. At this
depth, the thermal forcing becomes negative, inducing freezing at the ice shelf base. This process, with melt in the deep
grounding zone and freezing at intermediate depths, is called the ice pump effect. Observations of marine ice reveal a number
of regions with dominant basal freezing: north of the central ice rises, north of Fowler peninsula, and in the western boundary
regions of both the Filchner and Ronne Ice Shelves (Sandhäger et al., 2004; Holland and Feltham, 2005; Holland et al., 2007).
Both the 3D model as well as LADDIE qualitatively reproduce basal freezing in these regions (Fig. 8c,d). Quantitatively,
LADDIE simulates weaker refreezing north of the ice rises, which is partly caused by the idealised, uniform tidal forcing of
the model.

Further downstream, net melt rates are observed along the ice shelf front (Fig. 8b). These melt rates are reproduced by
the 3D model, but not by LADDIE. The local enhancement of melt along the ice shelf front is caused by the circulation of
warmer water within the cavity and by local peaks in tidal velocities (Makinson et al., 2011; Hausmann et al., 2020). The
highly idealised forcing of LADDIE lacks these spatial patterns, which at least partly explains the lack of simulated ice shelf
front melt in LADDIE. We note that a more detailed forcing, including 3D fields of ambient temperature and salinity, and a
non-uniform 2D field of tidal velocities, may improve the spatial patterns of basal melt and freezing by LADDIE. However,
we deem it to be a significant result that the qualitative large-scale patterns of basal melt and freezing can be reproduced using
simple 1D forcing fields and uniform tidal velocities.

### 3.2.2 Rutford grounding zone

Similar to Crosson-Dotson, the highest basal melt rates on the Filchner-Ronne Ice Shelf are found in the deep grounding zones.
One example of these is the Rutford Grounding Zone (Fig. 9). Compared to the Kohler Grounding Zone (Fig. 4b), the observed
basal melt rates in the Rutford Grounding Zone are of poorer quality (Fig. 9b). This observed basal melt field does not reflect
detailed channelised patterns that are clearly visible in the ice shelf topography (Fig. 9a). However, these observations do show
an overall pattern of high basal melt rates close to the deepest grounding line, and lower basal melt rates toward the shallower
regions. This pattern is confirmed by in-situ observations (Jenkins et al., 2006) as well as estimates from remote sensing (Rignot
et al., 2013; Moholdt et al., 2014).

These highest basal melt rates near the grounding line occur due to the low freezing temperature and the steep slope of the
ice shelf draft. The low freezing temperature enforces a relatively high thermal forcing. In addition, the steep slope facilitates
a rapid meltwater flow and a high turbulent exchange of heat toward the cold ice shelf base. Regarding the magnitude of
the basal melt rates near the grounding line, in-situ observations have revealed an overestimation of remote sensing-based
estimates (Jenkins et al., 2006). LADDIE largely reproduces the near-grounding line basal melt estimates by remote sensing
(approx. 5 m yr$^{-1}$), in-situ observations in the Rutford Grounding zone estimate basal melt rates of approx. 1 m yr$^{-1}$. The
basal melt rates by LADDIE nearby grounding lines are strongly tied to the parameter $D_{min}$. Tuning down this parameter may
reduce the overestimation by LADDIE in this region.



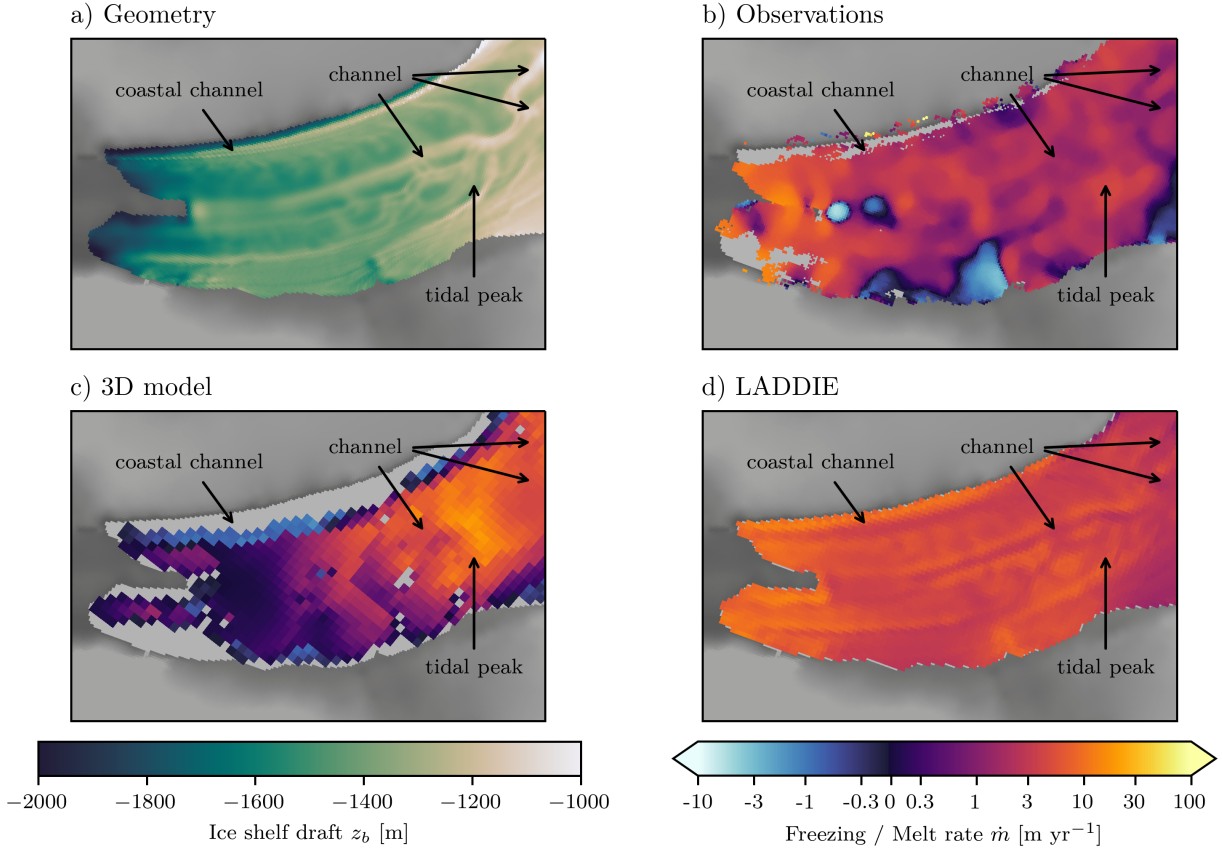

**Figure 9.** Rutford Grounding zone melt. Detailed features within the insets in Fig. 8.

In contrast to LADDIE, the 3D model NEMO simulates a spatial melt peak at the mouth of the grounding zone. This melt peak (Fig. 9c) is induced by the tidal forcing, which locally increases the thermal forcing (Hausmann et al., 2020). A similar tide-induced peak near the mouth of the grounding zone was simulated using a global setup of the 3D NEMO model (Mathiot et al., 2017). However, Padman et al. (2018) showed that this tidal peak is induced by errors in topography used in barotropic models to resolve tides. They further revealed that these tidal peaks are not in accordance with in situ tidal observations. To resolve this, they corrected for the water column thickness in the grounding zones, thereby removing the tidally induced melt peaks. After this topographic correction, their simulations of basal melt showed the highest basal melt rates in the Rutford grounding zone near the grounding line; this was the case with and without tidal forcing. In conclusion, we consider the simulated tidal peak (Fig. 9c) an unrealistic feature.

The topography of the Rutford grounding zone (Fig. 9a) contains a number of channels extending from the deepest grounding line to the mouth of the grounding zone. As revealed by Alley et al. (2016), the basal channels in the grounding zones of Filchner-Ronne are primarily subglacially sourced. This implies that subglacial meltwater forms these channels in the grounded parts of the glacier, and these channels are advected into the ice shelf as the ice begins to float. In contrast to the suspected ocean-





sourced basal melt channels of the Crosson-Dotson Ice Shelf, subglacially sourced channels can persist without enhanced basal
melt. The simulated enhanced melt rates along these channels, however, can aid their persistence and potentially contribute to
their westward migration as described in Sec. 3.1.3. In order to simulate subglacially sourced channels in a coupled setting,
LADDIE should be forced with subglacial meltwater.

The basal melt patterns simulated by LADDIE (Fig. 9d) propose an enhanced ice pump mechanism within these basal
channels. Near the grounding line, melt rates are increased within the channel, compared to outside the channel. This effect is
similar to the high melt rates in ocean-sourced basal channels; increased melt arises due to the convergence of meltwater plumes
and the consequential locally enhanced turbulent exchange. However, further downstream, toward the mouth of the grounding
zone, the ice shelf draft within the channels is shallower than outside these channels. Hence, the freezing temperature within
the channel is higher than outside, leading to a lower thermal forcing within the channel. Toward the grounding zone mouth,
basal melt within the channels is weaker than outside the channels; this contrast in basal melt rates implies an erosion effect of
basal melt on these subglacially sourced channels. We postulate that this erosion effect contributes to the absence of channels
in the shallower parts of the Filchner-Ronne Ice Shelf. Note that this erosion effect is not present in the small, warm ice shelves
like Crosson-Dotson, where ocean-sourced meltwater plumes extend to the ice shelf front. Throughout the whole length of
these ice shelves, basal melt rates are enhanced compared to outside the channels.

### 545  3.2.3  Western Boundary

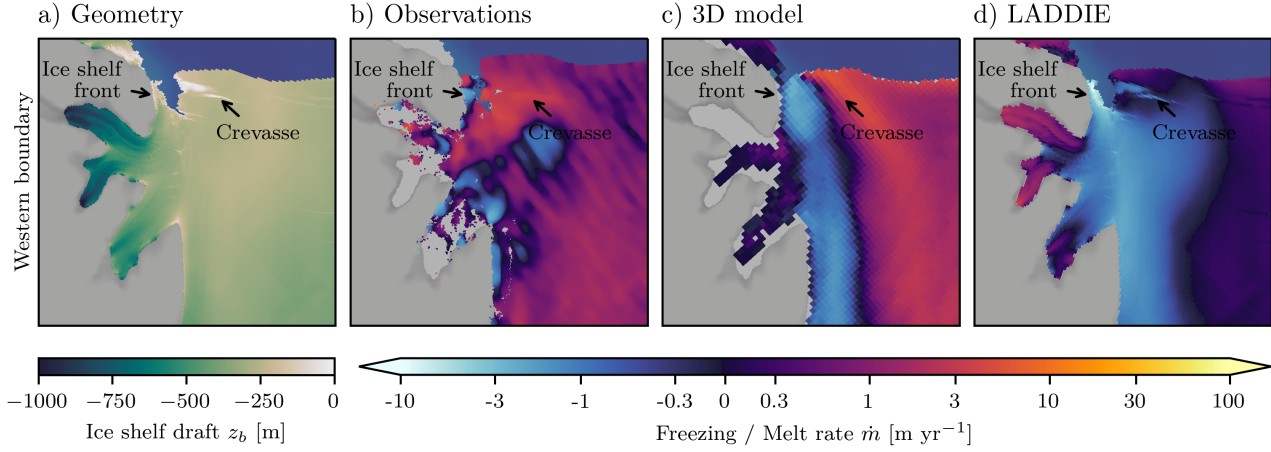

**Figure 10.** Western Boundary melt. Detailed features within the insets in Fig. 8.

Along both Western Boundaries of the Filchner and Ronne Ice Shelves, marine ice is detected (Sandhäger et al., 2004;
Holland and Feltham, 2005); this indicates basal freezing. And indeed, observations and model simulations reveal basal freez-
ing along these western boundaries (Fig. 8b,c,d). However, a discrepancy is visible toward the Ronne Ice Shelf front. Here,
observations indicate basal melting of the ice shelf (Fig. 10b), whilst both NEMO and LADDIE simulate basal freezing all

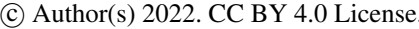


the way to the ice shelf front (Fig. 10c,d). As shown by simulations by Hausmann et al. (2020) and Padman et al. (2018), the presence or absence of tides cannot explain basal melt at this ice shelf front. In addition, the basal freezing is retained when the cavity circulation is explicitly solved in a 3D model, either regionally (Hausmann et al., 2020) or globally (Mathiot et al., 2017). In contrast, simulations that do resolve basal melt at this ice shelf front overestimate the extent of basal melt along both the western boundaries of the Ronne and Filchner ice shelves (Richter et al., 2022). The extent of basal melt and basal freezing along these western boundaries, and the underlying physics determining this extent, is therefore a remaining issue to be resolved by the wider ocean modeling community.

What is relevant for our current study is the consistency between 3D ocean models and LADDIE. LADDIE should be interpreted as a simplified model with less physics at a higher resolution compared to 3D ocean models. Hence, LADDIE is not a suitable model to solve this discrepancy between observed basal melt and simulated basal freezing. Rather, 3D ocean models should be used to assess the physics underlying this basal melt. Processes such as polynya formation and summer solar heating in front of an ice shelf (Stewart et al., 2019) can induce basal melt near ice shelf fronts. When simulated correctly using a 3D ocean model, this process will be reflected in the ambient temperature and salinity fields produced by that model. These fields can then be used to force the LADDIE model in order to check whether the basal freezing at the ice shelf front can be removed.

Near the ice shelf front of the Ronne Ice Shelf, a large crevasse exists (Fig. 10a). The observations show slightly elevated basal melt in this region (Fig. 10b). In contrast, previous modeling studies indicated net freezing in crevasses due to frazil ice formation (Jordan et al., 2014). The 3D model results do not resolve any spatial variations in basal melt or freezing at this scale (Fig. 10c). Within the crevasse, LADDIE simulates enhanced basal freezing. Here, two factors are at play. Within the crevasse, the pressure melting point is relatively high, creating a large negative thermal forcing. In addition, the steep slopes around the crevasse allow for a strong turbulent exchange. The latter factor can also explain the slightly elevated basal melt in the observations, creating an enhanced turbulent exchange in a region of positive thermal forcing. This is in line with the discrepancy described above, that the ambient temperatures near this ice shelf front are likely higher than those prescribed as forcing to LADDIE. Based on this, we argue that more realistic forcing fields could improve the simulated basal melt rates by LADDIE in this region near the Ronne Ice Shelf front.

## 4 Discussion

In the quest for a computationally efficient method to simulate physically plausible basal melt fields, we have presented the 2D model LADDIE. The model can be seen as a tool to simulate basal melt rates that sits between computationally cheap, yet physically more idealised parameterisations and physically more realistic, yet computationally heavy 3D ocean models. The relevance of LADDIE to the field of ice sheet modeling relies on its ability to combine the strengths of parameterisations and 3D ocean models. This means that the model must be computationally significantly cheaper than 3D models whilst producing significantly more realistic basal melt fields than idealised parameterisations.



In terms of its computational cost, LADDIE falls in between simple parameterisations and 3D ocean models. Basal melt parameterisations (e.g., Reese et al., 2018a; Jourdain et al., 2020) can be computed online with negligible computational cost. In contrast, 3D ocean models require considerable resources; for that reason, the resolution in the ISOMIP+ protocol was adjusted from $1\times1$ to $2\times2$ km$^2$. The higher resolution would pose a significant limitation to the participation of 3D ocean models (Asay-Davis et al., 2016). For LADDIE at a $1\times1$ km$^2$ resolution, a full spin up requires 0.5 CPU hours for the Crosson-Dotson Ice Shelf and 60 CPU hours for the Filchner-Ronne Ice Shelf. This allows for the configuration at a higher resolution, hence resolving more spatial detail. In addition, the difference in the computational cost makes LADDIE particularly suitable for extensive tuning and sensitivity experiments for which a large number of simulations are required.

The spatial detail of the basal melt fields simulated by LADDIE are of comparable quality to those produced by 3D ocean models. A particular improvement compared to basal melt parameterisations is that LADDIE resolves the influence of ice shelf topography and Coriolis deflection on the meltwater flow. This allows for the simulation of physically plausible enhanced melt rates in basal channels and shear zones, as well as the western intensification of basal melt plumes within basal channels. Each of these features can have a significant effect on ice shelf dynamics and thus on ice–ocean interactions. In comparison to 3D ocean models, LADDIE reproduces most basal melt features despite its idealised physical description. The higher resolution at which LADDIE can be configured allows for more detail in fine-scale features such as basal channels; again, the simulation of these kilometer-scale features may be important for coupled ice–ocean modeling. In addition, LADDIE resolves significant basal melt rates at the grounding line. These melt rates may be overestimated for the Filchner-Ronne Ice Shelf (Jenkins et al., 2006), yet are in line with the assessed conditions to reproduce historical grounding line retreat of the Crosson-Dotson Ice Shelf (Lilien et al., 2019).

In this study, the main simulations were based on idealised forcing, combining horizontally uniform profiles of temperature and salinity with horizontally uniform values of tidal forcing. As shown for the Crosson-Dotson Ice Shelf in App. A, quantitative improvements can be made to the simulated basal melt fields with 3D temperature and salinity forcing. In addition, spatially non-uniform tidal forcing may improve the basal melt and freezing patterns of the Filchner-Ronne Ice Shelf (Makinson et al., 2011; Hausmann et al., 2020). We caution though that incorrect tidal velocities in grounding zones can obscure melt rates in these regions, unless corrected for as described by Padman et al. (2018). The simulations were performed without subglacial outflow. This external forcing could easily be added as a volume source in Eq. 1 and may affect near-grounding line basal melt rates (Jenkins, 2011). In addition, in coupled ice–ocean settings including LADDIE, subglacial outflow is an essential process that forms topographic channels. For the study of the topographic impact on basal melt, however, we considered idealised forcing profiles to be more illustrative. Based on these, we conclude that a detailed geometry combined with reasonable forcing fields allows for the reproduction of the majority of basal melt patterns.

The model version presented here is the first version of the LADDIE model, which can be expanded in a number of ways. One missing aspect is the option for convection-driven melting in regions where shear-driven melting is absent. This process is inherently difficult to parameterise (Rosevear et al., 2022) and therefore currently not included in the model. Rather, melting in regions with low velocities is induced in LADDIE by enhancing local entrainment in order to ensure a minimial layer thickness $D_{min}$. This process can be replaced when a suitable parameterisation for convection-driven melting is developed. Similarly,





the impact of double diffusion on the vertical heat exchange (Rosevear et al., 2021) is not included in the present model version. Another process that is currently not included is the limitation imposed by the available water column thickness. In most regions, the plume thickness is significantly less than the total water column thickness. Therefore, no bathymetric constraint is imposed in the present version of LADDIE. However, the thin water column nearby grounding lines suppresses the horizontal transport of heat which is entrained into the meltwater layer. Limiting this heat transport through bathymetric constraints would reduce basal melt nearby grounding lines, possibly reducing the overstimation of basal melt rates in the Rutford grounding zone. Future model development and validation should combine the inclusion of bathymetric constraints with detailed validation of grounding zone melt to in-situ observations at various ice shelves. Finally, basal freezing commonly occurs due to frazil ice formation and deposition (Holland and Feltham, 2005; Jordan et al., 2014). The overall patterns of basal freezing by LADDIE appear physically plausible, yet these may be improved by explicitly simulating frazil ice formation. All of the above processes are considered possible expansions of the model, beyond the scope of this first model version.

Despite the computational advantages of a 2D model, this configuration poses a number of limitations to the applicability of LADDIE. The 2D configuration prevents the explicit computation of certain processes such as turbulence, barotropic flow, and conservation laws in the complete cavity. In addition, the model design makes LADDIE unsuitable for the study of a number of physical processes related to Antarctic ice shelf basal melt. First and foremost, LADDIE cannot resolve shelf processes governing the heat exchange between the deep ocean and the continental shelf (e.g., Thompson et al., 2018). In addition, the model cannot address questions related to time scales and tipping points related to the cavity circulation (Naughten et al., 2021). Finally, the effects of sea ice, such as solar absorption in polynyas (Stewart et al., 2019), are beyond the model scope. Research questions such as these require 3D ocean models and coupled ESMs to be studied. For such problems, LADDIE can only function to downscale basal melt rates based on output from 3D ocean models.

Both LADDIE and 3D ocean models are primarily validated by comparison to observations of basal melt. However, these observations are inherently uncertain. Most observations are indirectly derived from remote sensing estimates of altimetry. Their conversion to basal melt rates relies on estimates of the surface mass balance and the ice flow convergence, combined with the assumption of flotation. Uncertainties in each of these aspects reduce the reliability of estimated basal melt rates nearby grounding lines, in highly dynamical regions such as shear zones, and on small scales such as within kilometer-width channels. Due to these uncertainties, discrepancies between observations and model results do not unequivocally point to model deficiencies. Hence, the limited availability and reliability of observed basal melt rates is one bottleneck for the validation, tuning, and development of LADDIE and other 3D ocean models. The scarcity of observations, however, also creates the option to use LADDIE for applications of data assimilation and/or reanalysis.

# 5 Conclusions

In this study, we have introduced the basal melt model LADDIE. The model resolves the layer thickness, momentum, temperature, and salinity of the upper ocean layer below ice shelves on a 2D grid. The aim of the model is to simulate basal melt patterns of comparable quality to 3D ocean models, yet with a considerably lower computational cost which allows LADDIE





to be configured on a finer grid. We have validated the model by comparing simulated basal melt fields of the Crosson-Dotson and Filchner-Ronne Ice Shelves to observations and to 3D ocean model output. By resolving topographic steering and Coriolis deflection of the meltwater flow, LADDIE produces significantly more detailed basal melt patterns than parameterisations of lower order. LADDIE realistically simulates the ice shelf average melt rates as well as the large-scale pattern of basal melt and freezing in both warm and cold environments. In addition, LADDIE simulates physically plausible small-scale features includ-

ing basal melt within topographic channels, along shear margins, and nearby grounding lines. Its performance on small spatial scales makes the model particularly suitable for providing basal melt fields as forcing for high-resolution ice sheet models. We thus conclude that LADDIE can function to bridge the resolution gap between ocean models and ice sheet models.

*Code and data availability.* The model code is available on https://github.com/erwinlambert/laddie (and it will be permanently released on zenodo together with the data presented here once the publication is accepted). The BedMachine v2 data is available from https://nsidc.org/data/nsidc-
660 0756





## Appendix A: Sensitivity to forcing

The LADDIE model can be forced with either 1D profiles of ambient temperature and salinity, or 3D fields representing the cavity. In the main text of this paper, idealised 1D profiles are applied. In this section, we address the impact of 3D forcing. In Fig. A1, melt patterns under both forcing options are shown. The 1D forcing is identical to that described in the paper. The 3D
forcing is derived from the MITgcm model over the period 2003–2008.

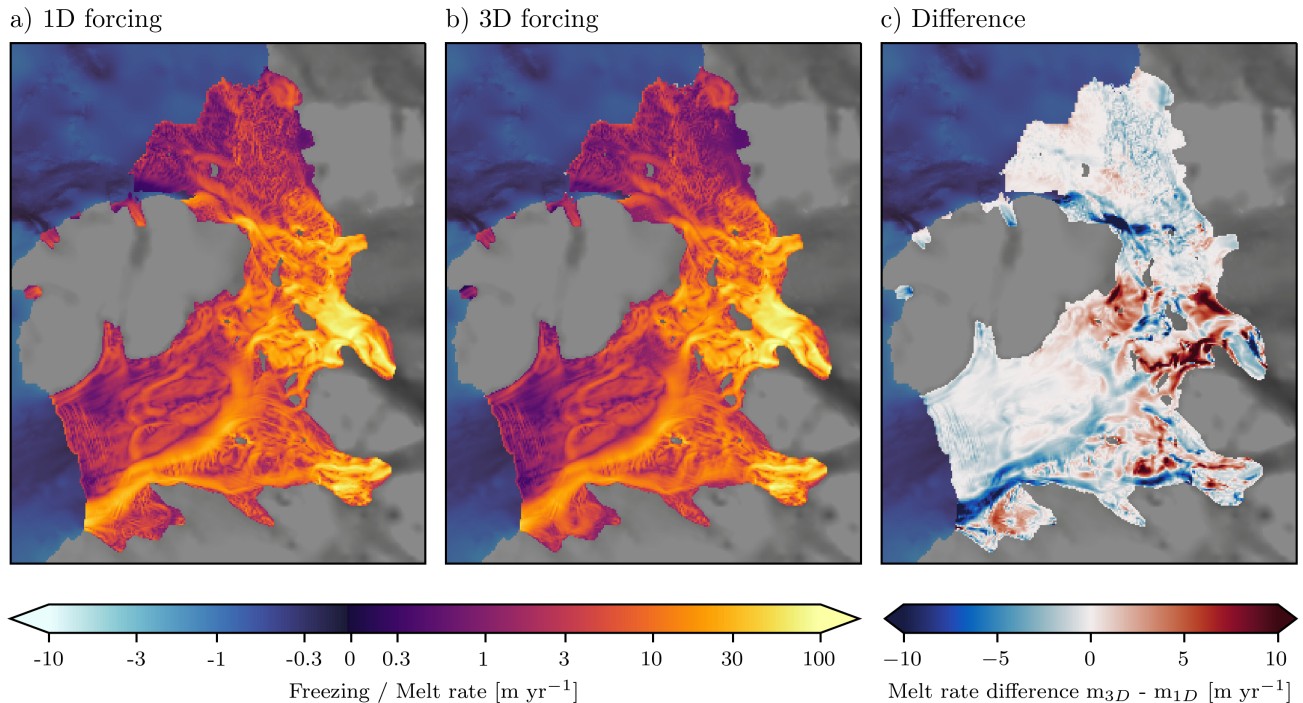

**Figure A1.** Impact of forcing on Crosson-Dotson melt rates. a) Melt rates from 1D forcing, identical to Fig. 3d. b) Melt rates from 3D forcing. c) Difference between a) and b).

Notably, the melt patterns (Fig. A1a,b) show a strong similarity. From large to small scales, melt patterns and plumes are resolved regardless of the forcing field. These patterns are governed by the topography which has a first-order impact on the basal melt patterns. The differences arise primarily in the high-melt region downstream from the Smith grounding zone (higher melt for 3D forcing) and the melt channels and shear zones (lower melt rates for 3D forcing). These differences arise due to
the differences in the applied forcing fields, including the stratification therein. We note that a greater overlap between the two cases could be achieved if the 1D forcing profiles were derived directly from the 3D MITgcm fields. However, this comparison shows the difference between the simplest application of the model (idealised 1D forcing) and the most complex one (detailed



3D forcing). Although significant differences exist, the basal melt patterns are nearly identical, implying that the exact forcing has a second order effect. A major exception to this is the average basal melt rate, which is strongly tied to the forcing applied.

## Appendix B: Tuning


As described in the main text, we consider two parameters, $C_{d,top}$ and $D_{min}$ as tuning parameters. The tuning is based on the Crosson-Dotson simulations with 3D forcing from MITgcm, averaged over the period 2003-2008. For both parameters, we apply a tuning procedure based on five tuning targets described below. The parameters are tuned for three horizontal resolutions, namely $\Delta x = 0.5$ km, 1.0 km, and 2.0 km. We scale the horizontal viscosity and diffusion linearly with the resolution, leading

to values $A_h = K_h = 25$, 50, and 100 m$^2$s$^{-1}$ respectively. These latter values were based on trial and error, largely guided by numerical stability.

For each resolution, 16 simulates are performed with all combinations of $C_{d,top} \in [0.8; 1.0; 1.2;$ and $1.4 \times 10^{-3}]$ and of $D_{min} \in [2; 4; 6; 8$ m]. For each of these 16 simulations, the diagnostics described below are quantified. Based on the best fit to the tuning targets, for each of the three resolutions, the optimal values of $C_{d,top}$ and $D_{min}$ are extracted.

### 685 B1 Tuning targets

As tuning targets, we consider a range of five diagnostics. The average melt rate over the total Crosson-Dotson Ice Shelf is tuned to a range of 8.7–9.9 m yr$^{-1}$. This range is based on the average melt rate of 9.3$\pm$0.6 m yr$^{-1}$ derived from remote sensing observations over the period 2003–2008 by Rignot et al. (2013).

The following three diagnostics represent the spatial melt pattern. The melt rate in the Kohler grounding zone, the Dotson

Channel, and the centerline along this Dotson Channel (see Fig. 3 for the locations). The melt rates in these regions are tuned to remote sensing observations over a different period (2010–2016). To compensate for interannual variability, we tune the model to melt amplifications (regional melt divided by ice-shelf average melt), rather than the actual melt rates. These amplifications were found to be more robust on an interannual basis.

For the Dotson Channel and the centerline along this channel, tuning ranges are derived from the remote sensing observations

over the period 2010–2016 by Gourmelen et al. (2017), equal to those presented in the main text. Melt amplifications are derived by dividing these regional melt rates by the Crosson-Dotson average melt rate of 6.89 m yr$^{-1}$ over 2010–2016 (Goldberg et al., 2019). Note that we follow Goldberg et al. (2019) and average melt rates by omitting negative values. This leads to regional melt amplifications in the Dotson Channel of 1.2 and along the Channel Centerline of 2.8. Without reported uncertainties in the data set, we are forced to prescribe arbitrary uncertainties. We apply tuning ranges of 1.1–1.3 and 2.4–3.2 respectively.

For the Kohler grounding zone, we combine estimates from the same remote sensing data set with estimates from radar observations over the comparable period 2010–2014 (Khazendar et al., 2016). The average melt rate over the Kohler grounding zone from remote sensing is 25.5 m yr$^{-1}$. From radar observations, this is 20.4 m yr$^{-1}$. Using these values as upper and lower bounds for the tuning range gives a range for the Kohler amplification of 3.0–3.8.





The fifth and last diagnostic used as a tuning target is the total overturning circulation. The overturning in LADDIE is

interpreted as the integrated entrainment of ambient water into the upper layer. This volume flux, together with the small
volume flux of meltwater, exits the meltwater layer through detrainment and outflow across the ice-shelf front. We extract
observations from the years 2006, 2007 and 2011 during which melt rates were comparable to the period 2003–2008 (Jenkins
et al., 2018). We take a tuning range based on the lowest and highest observations among these three years, leading to a range
of 0.39–0.61 Sv.

**B2    Tuning results**

The tuning results are shown in Fig. B1, with the first five columns representing the five tuning targets, and the three rows
representing the three resolutions. The coloured shading displays the simulated diagnostic values. The grey shaded zone in
each panel marks the tuning range. In the sixth column, the overlap of the five tuning ranges is shown, with darker regions
marking a better agreement with the tuning targets.

Based on this overlap, the optimal values for $C_{d,top}$ and $D_{min}$ are determined, marked by the yellow dots. For resolutions of
1.0 and 2.0 km, four of the five diagnostics fall within the tuning range, with the Channel Centerline (CC) being the exception.
For the resolution of 0.5 km, all five tuning targets are met and a full agreement was found between model and observations.

We find that we can choose the same optimal value for $C_{d,top}$ for each resolution. This value of $1.1 \times 10^{-3}$ is close to the
value of $1.0 \times 10^{-3}$, taken by previous studies (Jourdain et al., 2017; Mathiot et al., 2017). Keeping this same value of $C_{d,top}$

for all resolutions, we find that differences in resolution – and correspondingly the viscosity $A_h$ and diffusivity $K_h$ – can be
compensated by varying $D_{min}$. Keeping the same parameter values, the average melt and the Kohler amplification decrease
when the resolution is increased. Decreasing $D_{min}$ reduces the thickness of the thermal barrier that the simulated layer forms
between the warm ambient water and the ice shelf draft. The consequence is that a reduced value of $D_{min}$ increases the average
melt and the Kohler amplification, thus compensating for the increased resolution.

The single tuning target that cannot be obtained with all resolutions is the Channel Centerline (CC) amplification. The
observed range of 2.4-3.2 can only be reproduced using a resolution of 0.5 km (Fig. B1p). Whilst the melt amplification over
the full width of the Dotson Channel can be reproduced with each resolution, the sharpness of the melt peak across this channel
requires a high resolution and a low viscosity and diffusivity. This finding highlights the importance of high-resolution basal
melt patterns. Forcing an ice shelf model with basal melt rates at a too coarse resolution, may underestimate the thinning along

the centerline of the channel. This can lead to an underestimation of the time it takes for the channel to 'burn through' the ice
shelf (e.g., Gourmelen et al., 2017). The optimal parameter values for each resolution are used throughout the current study for
both Crosson-Dotson and Filchner-Ronne Ice Shelves.

In order to show the robustness of the tuning procedure to the interannual variability, we have performed simulations for
each year betwen 1979 and 2012, using annual mean 3D forcing from the MITgcm simulations (Fig. B2). For the average

melt (Fig. B2a), the interannual variability is strong and time periods must be selected carefully, as we have done. For the
regional melt applications (Fig. B2b-d) and the overturning circulation (Fig. B2e), the interannual variability is comparable to





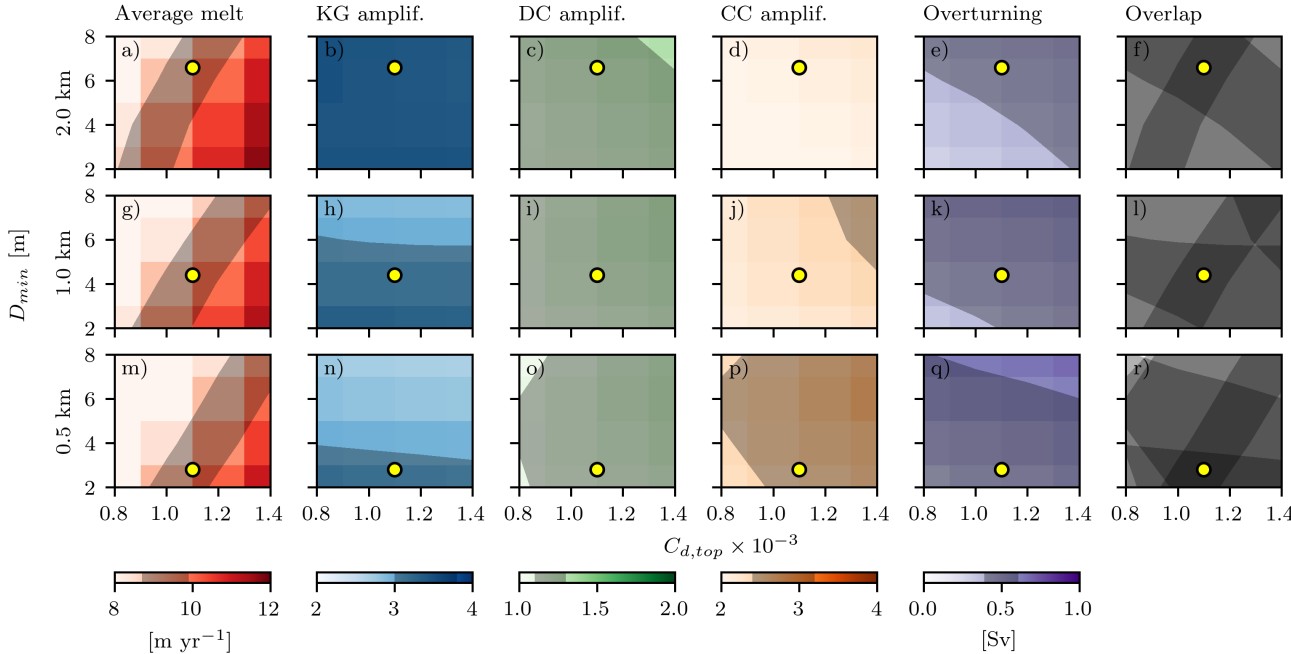

**Figure B1.** Parameter sensitivity test and tuning results. The three rows represent three horizontal resolutions (2.0, 1.0, and 0.5 km). Each column represents a diagnostic. Columns 2–4 denote the dimensionless melt amplification in the Kohler Grounding zone (KG), the Dotson Channel (DC) and along the Channel Centerline (CC). The grey shading denotes the tuning targets based on observations. The last column (Overlap) displays the overlap of these grey shaded regions, with darker colours indicating a better fit to observations. The yellow dots denote the chosen parameters for each resolution.

the assessed uncertainty. Hence, the exact time period used to gather the tuning targets from observations is less important. Based on this, we are confident that the above tuning procedure is robust with respect to interannual variability.

*Author contributions.* EL and AJ conceived the idea of the paper. EL wrote most of the code with significant contributions from AJ. PH
contributed to the model physics. All authors contributed to the scoping and writing of the paper.

*Competing interests.* The authors declare no competing interests.

*Acknowledgements.* EL was funded by the Netherlands Organization for Scientific Research (grant no. OCENW.GROOT.2019.091). AJ was funded by the Netherlands Polar Programme and the Water, Climate and Future Deltas program of Utrecht University. The authors thank U.

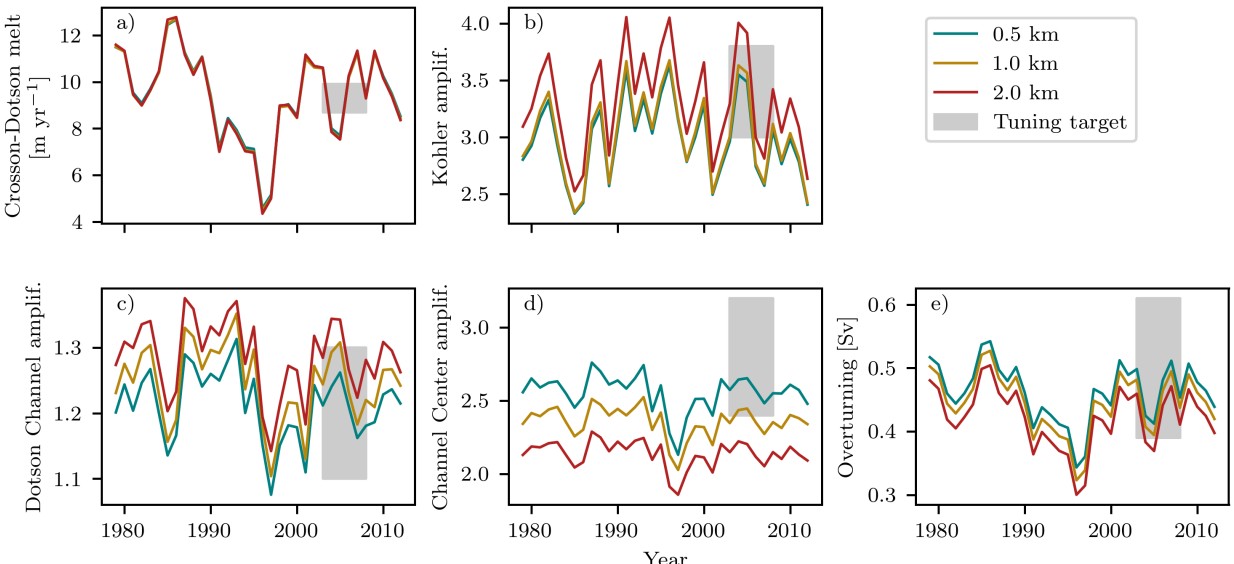

**Figure B2.** Time series of the main diagnostics for the Crosson-Dotson Ice Shelf (the same as in Fig. B1). For each year, the annual mean temperature and salinity fields from MITgcm are used to force LADDIE until steady-state from a motionless initial state. The coloured lines again denote the different model resolutions. The grey regions denote the tuning targets, based on observations, applied to the period 2003–2008.

Hausmann for sharing the NEMO output and N. Gourmelen for the remote sensing data of Crosson-Dotson. The authors are grateful to T.F.
Jesse for the extensive model testing.



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
