# Peer review of "Modeling Antarctic ice shelf basal melt patterns using the one-Layer Antarctic model for Dynamical Downscaling of Ice–ocean Exchanges (LADDIE v1.0)"

_The Cryosphere, 2022_

## Referee Comment (RC1)

Review for Lambert et al. :
**Modeling Antarctic ice shelf basal melt patterns using the one-Layer Antarctic model for Dynamical Downscaling of Ice–ocean Exchanges (LADDIE)**
Submitted to *The Cryosphere*

*Reviewer: Clara Burgard*
*I do this review un-anonymously to clarify from which background and level of expertise some remarks might come from and because it makes the conversation during review more transparent on both sides.*

**Summary**

The authors present the new simple model LADDIE that can be used (1) as a high-resolution parameterisation to link hydrographic properties in front of an ice shelf and melt at its base and (2) as a method to use information from coarse ocean models resolving the circulation in ice-shelf cavities to simulate high-resolution basal melt patterns. The authors present the model and its tuning (done on the Crosson-Dotson ice shelf) and then evaluate it on two ice shelves with different characteristics: Crosson-Dotson and Filchner-Ronne.

This model is an advancement compared to "classic" parameterisations in the sense that it includes 2D effects like the Coriolis force and provides the possibility to include fine-scale bathymetric characteristics in the resulting melt patterns. The topic is timely as the representation of basal melt in models remains a large source of uncertainty for future Antarctic ice-sheet projections. In particular, LADDIE enables the resolution of fine-scale channels and regions near the grounding line, where high melt occurs, and which are therefore crucial when forcing ice-sheet models. Its application therefore has the potential to improve the forcing of ice-sheet simulations.

The manuscript is pleasant to read and the procedure to set up and evaluate the model is thoroughly described. I am curious to see how the application of LADDIE will change the behaviour of ice-sheet simulations when it will be ready for a more widespread use!

Before publication, however, I think that a few points need to be addressed to clarify this manuscript and make it more robust, especially concerning the evaluation procedure. I hope it is only a matter of restructuring and reformulating and does not involve redoing a major part of the analysis. I realise there are a lot of remarks but they come from sincere interest in the study. I hope that the authors can use them constructively and am looking forward to reading a clearer revised manuscript!

**GENERAL COMMENTS**

**General messaging**

One of the main confusion sources when I finished to read the manuscript was that it was not completely clear to me where we stand with LADDIE in the end. Is this manuscript a proof of concept, showing us that it works on two distinct ice-shelf categories or is it the presentation of a model that can be used directly now by ocean

and ice-sheet modelers? I recommend that the authors clarify this more in the introduction.

In particular, Table 2 nicely summarizes the tuned parameter for the two ice shelves considered. However, it is not clear from reading the manuscript which parameters to use when applying LADDIE to other ice shelves. Is retuning needed every time? I suggest to either start the manuscript with the clear message of "This is a feasibility study and we show it is possible" or discussing in the Discussion section how to decide on the parameters to use when applying LADDIE to other ice shelves.

**Tuning and evaluation setup**

LADDIE is tuned on one ice shelf, the Crosson-Dotson ice shelf, and then evaluated on this same ice shelf and another one (Filchner-Ronne - FRIS). In my opinion, this evaluation approach biases the results of the evaluation. Using the ice shelf used to tune as 50% of the evaluation is not necessarily robust. There is a high-resolution observational melting pattern available from Shean et al. 2019 and input profiles (Dutrieux et al. 2014) for Pine Island Glacier. I strongly suggest that the authors consider using this ice shelf to replace Crosson-Dotson either in the tuning or evaluation.

In addition, the tuning was done using idealized profiles as input and then comparing to a number of remote sensing and in-situ observational sources". I wonder why the tuning was not done using observational input directly, instead of idealized profiles. This would increase the consistency between "what comes in" and "what comes out". Can the authors clarify their reasoning?

**Evaluation setup**

The authors use several types of temperature and salinity profiles/fields as input to LADDIE: (1) idealized 1D profiles in front of the ice shelf and (2) spatial fields from ocean simulations. They then compare the resulting melt patterns to (1) observational estimates from satellites, (2) simulations from MITgcm/NEMO and (3) sometimes to "reasonably expected" patterns. Also, sometimes they compare the simulations from MITgcm/NEMO to observations.

I found it very confusing and difficult to read the evaluation and to follow it objectively because the patterns are cross-compared (obs to simulation, LADDIE to obs, LADDIE to simulations, LADDIE to reasonable expectations when obs and models were uncertain) in an order that, very crudely said, sometimes felt like "we choose to compare what suits us the most". I am sure that there was a more systematic approach than that in the comparisons but it is not necessarily clear yet unfortunately. I suggest the authors restructure these comparisons to clarify why they choose one comparison over another and why their way of doing is robust.

Another aspect that puzzled me regarding the evaluation is why 1D idealized profiles are used to then compare melt patterns to observational estimates. Why not use either the measured (Crosson-Dotson) and simulated (FRIS) profiles directly to make the evaluation more consistent?

Out of curiosity, have the authors compared the output of LADDIE run at the same resolution than the 3D models (using 1D input) and the output melt of the 3D models? This might be a good first step to evaluate LADDIE's large-scale patterns as a sanity check. I do not expect this to be done if the authors do not have it ready but it would be interesting to include if it had been done.

**Large VS small ice shelves**

The current version of LADDIE is computationally demanding for large ice shelves and 1D input profiles lead to larger uncertainties in the pattern. I suggest stressing in the conclusion that LADDIE is easier applicable to small ice shelves, especially with the 1D profile input, like classic parameterisations.

**"Validation" VS "Evaluation"**

The authors decide to use the word "validation" for their model. Personally, I have a strong opinion about the wording, and "validation" is very strong. In my opinion, a model can only be "evaluated" to understand if it does the things reasonably that we expect from it. As we know, "all models are wrong, but some are useful" and therefore "validating" seems a strong word for something we know is wrong. This is a personal opinion and I let the authors judge if they agree but I suggest the authors use "evaluation" instead of "validation".

**DETAILED COMMENTS**

**L4**: I suggest adding "on long timescales" after "this resolution"

**L6:** Could the authors define "offshore" in the manuscript. I would suggest calling this "far-field". My understanding of "offshore" is much further from the ice shelf, i.e. sometimes even further than the continental shelf, as is given by some coarse CMIP-type models. In this manuscript, offshore seems to stand for a region in front of the ice shelf, on the continental shelf. It is a fine but important distinction for the choice of input temperature and salinity, so I would appreciate if the authors could clarify the wording.

**L20:** I suggest leaving out "in particular those beyond 2100". There is enough uncertainty before 2100 already.

**L70:** I suggest adding "near Antarctica" or "at a latitude of XX°S" after 7.5 km because the km-resolution depends on where we are on the globe.

**L79:** add "e.g." before Favier et al., 2019

**Eq(5) and Eq(9):** missing the term containing $\dot{m}*S_i$. I suspect this is probably because $S_i$ is assumed to be 0. It might be worth mentioning this somewhere?

**L139:** Can the authors add a few words on what the reduced gravity stands for and why it is needed?

**Eq(7):** Define alpha and beta in the text.

**L146:** To avoid confusion, I suggest moving "below sea level" into the parenthesis after the zero.

**Section 2.1.1.:** \gamma_T, K_H, A_H are only introduced in a later section although they appear in Eq.1-5. I suggest that the authors already introduce them here. At least say what they stand for.

**L156:** Remove "." In front of citations

**Eq(10):** Should this be $z_b$?

**L160:** c_p and the lambdas should be defined here.

**L166:** In Jenkins (1991), \nu is called kinetic viscosity. Should \nu_0 be kinetic or molecular viscosity then? Same for the Prandtl number, it is defined as the "molecular" Prandtl number in Jenkins (1991).
On this, I admit that I am no expert but I just wanted to make sure these differences were no mistakes.

**L166:** I suggest adding "of seawater" in the end of the sentence

**L173:** Use \citep[e.g.][]{} for the citation

**Eq(14):** Define \mu. Also, I researched where this formula comes from. In my understanding, this might come from Eq. 35 from Gaspar (1988) but why is it missing the last term (the one with E_m^{3/2})?

**L183-185:** I suggest combining these two sentences because the first sentence does not read well as it is unclear between which terms there is a balance.

**L191, 203**: Gurvan et al. should be Madec et al. Gurvan is the first name.

**Table 1:** add "coefficient" for alpha and beta descriptions, again should \nu_0 be the "molecular" viscosity?

**L208:** I am not an expert on this particular concept but the problem of the transient state sounds like a serious limitation. Can the authors mention this again in the discussion and discuss a bit more how to reassess the parameter choice in future studies?

**L216:** "we have used" => "we use"

**L235:** I suggest replacing "reasonable" by "plausible"

**L241:** I suggest replacing "stressing that" by "although we are aware that"

**L243:** add "and" after "(2017)"

**L260:** To add one more layer of evaluation and to be consistent with FRIS, would it make sense to use the Adusumilli data for Crosson-Dotson as well?

**L267:** "shouthernmost" => "southernmost"

**Figure 2:** If possible, I suggest to move the colorbar to a more intuitive place

**Section 2.3.1:** This section is very similar to Section 2.2.1. I suggest combining them in the place it makes most sense. Maybe make Sec. 2.3.1 shorter with the information really focused on the model setup.

**L300:** As mentioned in the "General Comments", I do not follow why the authors use idealized forcing fields and then compare the resulting melt to observations. In particular, for Crosson-Dotson there are measurements, as the authors say themselves in L316. Using them would make the comparison more consistent and robust in my opinion.

**L304:** It is not clear to me why the authors do not directly use the fields from Padman et al. 2018 and choose to stay with constant U_tide. I would think that it would give insight into the effect of tidal velocity on the resulting melt. Can this choice be clarified?

**L330:** Here also, why not use the simulated profiles from Holland et al. 2007 directly?

**L337:** Move the definition of A_h and K_h to earlier

**L375:** I do not see clear patterns corresponding to the 450-m isobath in Figure 3b, sorry… They are clearer in Fig3c and d, as said in L382. However, I am not sure that this should be the main feature used to evaluate LADDIE if this pattern is not that clear in observations.

**L376:** When talking about the thermocline, it would make sense to also point to Fig. 2.

**L394-395:** Here, the authors acknowledge that a feature is good because it was tuned to it. In my opinion, this highlights the ambiguity of evaluating LADDIE in a detailed manner on an ice shelf and quantity it was tuned to.

**L397-404:** This paragraph is very confusing and difficult to follow. The authors state that the observations are not reliable near the grounding line and turn to numerical models to evaluate LADDIE. They then also evaluate the melt in MITgcm and compare the melt between MITgcm and LADDIE. This is what I meant in "General comments" when I meant that it is difficult to follow the different levels of evaluation. This paragraph is mixing too much. Can the authors restructure and clarify?

**L416:** About "the western side of the topographic channel": When talking North-East-South-West in Antarctica, it often gets confusing. Can the authors add labels for degrees East/West on the maps? Or they could add labels in the figure showing the "eastern flank" and "the western flank". That might make it easier to follow where to find the western side.

**Figure 6:** I suggest annotating "atmosphere", "ice", "ocean" directly in at least one panel to make the figure better readable intuitively.

**L425:** Would it make sense to add the results from MITgcm in Fig. 6 for a direct comparison and following better this sentence?

**L426-444:** These two paragraphs are a description and interpretation of features only resolved in LADDIE. I find these paragraphs difficult to interpret. LADDIE is used to make conclusions on processes, while it is being evaluated at the same time. Can we use it to interpret physical processes if we are not finished with the evaluation?

**L464-469:** Again, what I find difficult here is the mixing between evaluation and interpretation, making it difficult to follow. I suggest to better separate them, for example: (1) evaluation, (2) what do we learn from LADDIE.

**L485:** There is one "freezing" too much

**L491-492:** Is this a suggestion or have the authors looked at this into detail? If the latter, could the authors add one sentence how they came to this conclusion?

**L496:** I suggest that this lack of spatial pattern comes, at least partially, from the use of 1D profiles as input, which the authors say in the following sentence. However, earlier, the authors suggested that using a 1D profile is acceptable compared to 3D fields. I think this shows that this assumption holds on small ice shelves, like Crosson-Dotson, but that, on large ice shelves, 1D profiles introduce more uncertainty than 3D fields.

**L449:** I do not completely agree with the conclusion that the qualitative large-scale pattern can be reproduced. It is good near the grounding line, yes, but the missing melting at the front, in the West, and at the southern tip of Berkner Island. I would suggest to acknowledge more that limitations remain.

**L515:** Is it the goal to retune the parameter often? Again, this comes back to: Are the authors presenting a model to be used out of a box or is this more a proof of concept?

**L557-564:** This paragraph is confusing. I suggest reformulating or restructuring.

**L573:** "more realistic forcing fields" => I again do not understand why realistic forcing fields were not used directly, especially if the authors suggest that it would have given better results.

**Discussion:** If the authors do not plan to change their tuning evaluation setup, I suggest a clear and robust discussion paragraph about the influence of tuning and evaluating on the same ice shelf on the conclusions of the study.

**L581:** I suggest replacing "idealised" with "existing"

**L583:** The list of citations is not very exhaustive here. Either add "e.g." in front or also include Lazeroms et al. 2018 and 2019, Favier et al. 2019, Beckmann and Goosse, 2002.

**L591:** "compared to basal melt parameterisations" => hmm, ok. But the authors have not compared the LADDIE results to "classic" parameterisations before in the manuscript. Maybe add one or more citation where to find the patterns. Burgard et al. 2022 would be a possibility.

**L599:** Maybe because LADDIE was tuned towards reproducing Crosson-Dotson?

**L637-645:** Yes, the conclusions sound plausible but the paragraph is confusing. I suggest reformulating. In the end of this paragraph, I am left confused again about what was the precise goal of this study. I suggest the authors re-affirm in the beginning of the paper if the goal is a quantitative and/or qualitative evaluation. Due to the different levels of comparison, I was a bit lost.

**Discussion:** It would also help to add a paragraph about the planned application of LADDIE in the future. Would it be used for offline use or within an ocean model directly?

**L670-671:** I agree. And would it not be also more consistent?

**L674:** Can the authors do a similar comparison for FRISP? This would show how much of an influence the 1D vs 3D field has on a larger ice shelf.

**L682:** "simulates" => "simulations"

**L732:** Using the same parameters for both might be optimistic. Can the authors show similar figures for FRIS, just to compare, or does this involve tremendous effort?

---

## Author Comment (AC2)

Review for Lambert et al. :
Modeling Antarctic ice shelf basal melt patterns using the one-Layer Antarctic
model for Dynamical Downscaling of Ice–ocean Exchanges (LADDIE)
Submitted to The Cryosphere
Reviewer: Clara Burgard
I do this review un-anonymously to clarify from which background and level of expertise
some remarks might come from and because it makes the conversation during review
more transparent on both sides.

Summary

The authors present the new simple model LADDIE that can be used (1) as a highresolution
parameterisation to link hydrographic properties in front of an ice shelf and
melt at its base and (2) as a method to use information from coarse ocean models
resolving the circulation in ice-shelf cavities to simulate high-resolution basal melt
patterns. The authors present the model and its tuning (done on the Crosson-Dotson ice
shelf) and then evaluate it on two ice shelves with different characteristics: Crosson-
Dotson and Filchner-Ronne.

This model is an advancement compared to "classic" parameterisations in the sense that
it includes 2D effects like the Coriolis force and provides the possibility to include finescale
bathymetric characteristics in the resulting melt patterns. The topic is timely as
the representation of basal melt in models remains a large source of uncertainty for
future Antarctic ice-sheet projections. In particular, LADDIE enables the resolution of
fine-scale channels and regions near the grounding line, where high melt occurs, and
which are therefore crucial when forcing ice-sheet models. Its application therefore has
the potential to improve the forcing of ice-sheet simulations.

The manuscript is pleasant to read and the procedure to set up and evaluate the model
is thoroughly described. I am curious to see how the application of LADDIE will change
the behaviour of ice-sheet simulations when it will be ready for a more widespread use!

Before publication, however, I think that a few points need to be addressed to clarify
this manuscript and make it more robust, especially concerning the evaluation
procedure. I hope it is only a matter of restructuring and reformulating and does not
involve redoing a major part of the analysis. I realise there are a lot of remarks but they
come from sincere interest in the study. I hope that the authors can use them
constructively and am looking forward to reading a clearer revised manuscript!

We appreciate the reviewer's thorough and constructive remarks. Besides minor changes, we
will reorganise the Results section and respecify the study aims following the suggestions and
concerns of the reviewer. Below, we provide a point-by-point reply and describe the changes
we propose to make to the manuscript.

GENERAL COMMENTS

General messaging

One of the main confusion sources when I finished to read the manuscript was that it was not completely clear to me where we stand with LADDIE in the end. Is this manuscript a proof of concept, showing us that it works on two distinct ice-shelf categories or is it the presentation of a model that can be used directly now by ocean and ice-sheet modelers? I recommend that the authors clarify this more in the introduction.

We acknowledge that this point was not clear. In short, the manuscript aims to do both: present a free-to-use model and provide a proof of concept based on two ice shelves. We have worked on restructuring the code to make it easier to run the model without prior knowledge, and will write out a proper README to include in the open source repository. In the abstract and summary, we will clarify more clearly that this model is available for use for ocean- and ice sheet modelers.

In particular, Table 2 nicely summarizes the tuned parameter for the two ice shelves considered. However, it is not clear from reading the manuscript which parameters to use when applying LADDIE to other ice shelves. Is retuning needed every time? I suggest to either start the manuscript with the clear message of "This is a feasibility study and we show it is possible" or discussing in the Discussion section how to decide on the parameters to use when applying LADDIE to other ice shelves.

Our evaluation of FRIS, using the tuning of Crosson-Dotson, was included as confirmation that a single tuning can suffice for simulating all Antarctic ice shelves. However, retuning can be useful / necessary in use cases where substantial observational data is available of either melt patterns or oceanic forcing. Also, retuning may be useful when forcing the model with biased input. We will include a paragraph in the Discussion devoted to describing recommendations for other ice shelves.

Tuning and evaluation setup

LADDIE is tuned on one ice shelf, the Crosson-Dotson ice shelf, and then evaluated on this same ice shelf and another one (Filchner-Ronne - FRIS). In my opinion, this evaluation approach biases the results of the evaluation. Using the ice shelf used to tune as 50% of the evaluation is not necessarily robust. There is a high-resolution observational melting pattern available from Shean et al. 2019 and input profiles (Dutrieux et al. 2014) for Pine Island Glacier. I strongly suggest that the authors consider using this ice shelf to replace Crosson-Dotson either in the tuning or evaluation.

Indeed, tuning and evaluation is combined in the Crosson-Dotson simulations. As we use only two tuning parameters for five diagnostics, we consider the evaluation to still be robust. However, we recognise that this procedure can raise questions on the robustness. Hence, we will follow your recommendation to include simulations for the Pine Island ice shelf, compared to the observations from Shean et al 2019 (See figure below, to be included in the appendix). As we prefer to keep the discussion of Crosson-Dotson in the main text, and to retain a good balance between the discussion of warm and cold ice shelves in the main text, we decided to include the Pine Island simulation in the appendix.

[Figure]

a) Geometry | b) Observations | c) 3D model | d) LADDIE

−1500 −1000 −500 0
Draft [m]

-10 -3 -1 -0.3 0 0.3 1 3 10 30 100
Freezing / Melt rate $\dot{m}$ [m yr$^{-1}$]

In addition, the tuning was done using idealized profiles as input and then comparing to a number of remote sensing and in-situ observational sources". I wonder why the tuning was not done using observational input directly, instead of idealized profiles. This would increase the consistency between "what comes in" and "what comes out". Can the authors clarify their reasoning?

In fact, the tuning was done using 3D model output, as we consider this to be closest to a realistic forcing field (and hence to provide the best comparison to observations). To avoid confusion, we will move the complete discussion on 3D forcing and tuning to the appendix.

Evaluation setup

The authors use several types of temperature and salinity profiles/fields as input to LADDIE: (1) idealized 1D profiles in front of the ice shelf and (2) spatial fields from ocean simulations. They then compare the resulting melt patterns to (1) observational estimates from satellites, (2) simulations from MITgcm/NEMO and (3) sometimes to "reasonably expected" patterns. Also, sometimes they compare the simulations from MITgcm/NEMO to observations.

I found it very confusing and difficult to read the evaluation and to follow it objectively because the patterns are cross-compared (obs to simulation, LADDIE to obs, LADDIE to simulations, LADDIE to reasonable expectations when obs and models were uncertain) in an order that, very crudely said, sometimes felt like "we choose to compare what suits us the most". I am sure that there was a more systematic approach than that in the comparisons but it is not necessarily clear yet unfortunately. I suggest the authors restructure these comparisons to clarify why they choose one comparison over another and why their way of doing is robust.

We acknowledge and agree that the presentation of the evaluation was poorly structured. We will restructure the Results section into three subsections per ice shelf:
- Model evaluation (to observations)
- Inter-model comparison (to 3D ocean models)
- New features (which cannot be evaluated by either observations or 3D models)
To align better with this new structure, we will condense the spatial melt patterns of each ice shelf, including the regional patterns, into a single figure.

Another aspect that puzzled me regarding the evaluation is why 1D idealized profiles

are used to then compare melt patterns to observational estimates. Why not use either the measured (Crosson-Dotson) and simulated (FRIS) profiles directly to make the evaluation more consistent?

Observed profiles are subject to strong interannual variability, hence they may not be representative for the evaluation. The idealised forcing is used to provide a clearer way to evaluate the simulated melt patterns. We will argue for this methodology more explicitly in the methods section.

Out of curiosity, have the authors compared the output of LADDIE run at the same resolution than the 3D models (using 1D input) and the output melt of the 3D models? This might be a good first step to evaluate LADDIE's large-scale patterns as a sanity check. I do not expect this to be done if the authors do not have it ready but it would be interesting to include if it had been done.

These experiments were indeed performed, and included in an earlier draft of the manuscript. As they provided negligible additional information (the melt patterns are similar, but less detailed than the higher resolution simulations), we decided to remove these. We believe that the figures in Appendix B contain sufficient information on the impact of spatial resolution for LADDIE. We will describe this in the revised paper.

Large VS small ice shelves

The current version of LADDIE is computationally demanding for large ice shelves and 1D input profiles lead to larger uncertainties in the pattern. I suggest stressing in the conclusion that LADDIE is easier applicable to small ice shelves, especially with the 1D profile input, like classic parameterisations.

This is a good point and we will include this in the discussion

"Validation" VS "Evaluation"

The authors decide to use the word "validation" for their model. Personally, I have a strong opinion about the wording, and "validation" is very strong. In my opinion, a model can only be "evaluated" to understand if it does the things reasonably that we expect from it. As we know, "all models are wrong, but some are useful" and therefore "validating" seems a strong word for something we know is wrong. This is a personal opinion and I let the authors judge if they agree but I suggest the authors use "evaluation" instead of "validation".

We agree and will replace 'validation' by 'evaluation' throughout the manuscript.

DETAILED COMMENTS

L4: I suggest adding "on long timescales" after "this resolution"

Agreed and will implement

L6: Could the authors define "offshore" in the manuscript. I would suggest calling this "far-field". My understanding of "offshore" is much further from the ice shelf, i.e.

sometimes even further than the continental shelf, as is given by some coarse CMIP-type models. In this manuscript, offshore seems to stand for a region in front of the ice shelf, on the continental shelf. It is a fine but important distinction for the choice of input temperature and salinity, so I would appreciate if the authors could clarify the wording.

We agree that this term is ambiguous and will mention this in the discussion where future use cases are discussed

L20: I suggest leaving out "in particular those beyond 2100". There is enough uncertainty before 2100 already.

Agreed and will implement

L70: I suggest adding "near Antarctica" or "at a latitude of XX°S" after 7.5 km because the km-resolution depends on where we are on the globe.

Agreed and will implement

L79: add "e.g." before Favier et al., 2019

Agreed and will implement

Eq(5) and Eq(9): missing the term containing $m^{\dot}*S_i$. I suspect this is probably because $S_i$ is assumed to be 0. It might be worth mentioning this somewhere?

This is correct, we will mention this in Sec. 2.1.1

L139: Can the authors add a few words on what the reduced gravity stands for and why it is needed?

Agreed and will implement

Eq(7): Define alpha and beta in the text.

Agreed and will do

L146: To avoid confusion, I suggest moving "below sea level" into the parenthesis after the zero.

We will change this to 'meters below sea level' to make it clearer

Section 2.1.1.: $\gamma_T, K_H, A_H$ are only introduced in a later section although they appear in Eq.1-5. I suggest that the authors already introduce them here. At least say what they stand for.

Agreed and will implement

L156: Remove "." In front of citations

Agreed and will do

Eq(10): Should this be $z_b$?

Correct, thanks. Will change

L160: c_p and the lambdas should be defined here.

Agreed and will implement

L166: In Jenkins (1991), \nu is called kinetic viscosity. Should \nu_0 be kinetic or molecular viscosity then? Same for the Prandtl number, it is defined as the "molecular" Prandtl number in Jenkins (1991).
On this, I admit that I am no expert but I just wanted to make sure these differences were no mistakes.

To avoid confusion, we will adopt the nomenclature of Jenkins (1991)

L166: I suggest adding "of seawater" in the end of the sentence

Agreed and will implement

L173: Use \citep[e.g.][]{} for the citation

Will change this to \citet

Eq(14): Define \mu. Also, I researched where this formula comes from. In my understanding, this might come from Eq. 35 from Gaspar (1988) but why is it missing the last term (the one with E_m^{3/2})?

This is the TKE dissipation term, to which Gaspar (1988) devotes much discussion. In our formulation, this dissipation is included within the mechanical production of TKE (the RHS of our equation (14)) and so does not appear separately. The assumption is that dissipation is simply a constant fraction of TKE production. This assumption originates with Niiler and Kraus (1977) and is described in equations (25)-(27) of Gaspar (1988). (Note that Gaspar's equation (25) also includes turbulence production by unstable convection, which is zero beneath melting ice shelves, and almost certainly negligible beneath freezing ice shelves.) Given the absence of detailed observations beneath ice shelves, we don't believe there is any basis for adopting a more complex formulation of the dissipation. We will summarise this discussion in the revised paper.

L183-185: I suggest combining these two sentences because the first sentence does not read well as it is unclear between which terms there is a balance.

Agreed, we will merge these sentences

L191, 203: Gurvan et al. should be Madec et al. Gurvan is the first name.

Thanks, we will correct this citation

Table 1: add "coefficient" for alpha and beta descriptions, again should \nu_0 be the

"molecular" viscosity?

Agreed and will implement

L208: I am not an expert on this particular concept but the problem of the transient state sounds like a serious limitation. Can the authors mention this again in the discussion and discuss a bit more how to reassess the parameter choice in future studies?

As we describe here, this is a limitation of the chosen configuration, not of the model itself. We will mention this aspect again in the discussion when presenting future use cases.

L216: "we have used" => "we use"

Agreed and will implement

L235: I suggest replacing "reasonable" by "plausible"

Agreed and will implement

L241: I suggest replacing "stressing that" by "although we are aware that"

We primarily want the reader to realise this, rather than mentioning that we are aware of it. Hence, we will keep the current formulation.

L243: add "and" after "(2017)"

Agreed and will implement

L260: To add one more layer of evaluation and to be consistent with FRIS, would it make sense to use the Adusumilli data for Crosson-Dotson as well?

We consider the Adusumilli data of substantially lower accuracy. This is a trade-off between consistency and quality, and we have opted for the latter. Unfortunately, we are not aware of any more reliable remote sensing estimates for FRIS.

L267: "shouthernmost" => "southernmost"

Thanks, we will correct this

Figure 2: If possible, I suggest to move the colorbar to a more intuitive place

Yes, we will revise the alignment of this figure

Section 2.3.1: This section is very similar to Section 2.2.1. I suggest combining them in the place it makes most sense. Maybe make Sec. 2.3.1 shorter with the information really focused on the model setup.

Agreed, we will minimise 2.3.1 to only describe the geometric setup for the specific used cases

L300: As mentioned in the "General Comments", I do not follow why the authors use idealized forcing fields and then compare the resulting melt to observations. In particular, for Crosson-Dotson there are measurements, as the authors say themselves in L316. Using them would make the comparison more consistent and robust in my opinion.

As argued above, the observations are subject to substantial interannual variability, and hence not representative of multi-year averages. Hence, we will keep the idealised forcing and will clarify our choice more explicitly.

L304: It is not clear to me why the authors do not directly use the fields from Padman et al. 2018 and choose to stay with constant U_tide. I would think that it would give insight into the effect of tidal velocity on the resulting melt. Can this choice be clarified?

The reason is two-fold. First, we were unable to obtain these data from the authors. Second, in line with the argumentation for idealised forcing profiles, we consider it valuable to present the model behaviour under simple forcing. We will phrase this argumentation more clearly in the methods.

L330: Here also, why not use the simulated profiles from Holland et al. 2007 directly?

See argumentation above. To assure the reader that the exact forcing is of less impact on melt patterns than the geometry, we have included appendix A

L337: Move the definition of A_h and K_h to earlier

Agreed and will do

L375: I do not see clear patterns corresponding to the 450-m isobath in Figure 3b, sorry… They are clearer in Fig3c and d, as said in L382. However, I am not sure that this should be the main feature used to evaluate LADDIE if this pattern is not that clear in observations.

Agreed, we will remove the contour line

L376: When talking about the thermocline, it would make sense to also point to Fig. 2.

Good idea, will implement

L394-395: Here, the authors acknowledge that a feature is good because it was tuned to it. In my opinion, this highlights the ambiguity of evaluating LADDIE in a detailed manner on an ice shelf and quantity it was tuned to.

We understand that this raises questions and refer the reviewer to our previous response

L397-404: This paragraph is very confusing and difficult to follow. The authors state that the observations are not reliable near the grounding line and turn to numerical models to evaluate LADDIE. They then also evaluate the melt in MITgcm and compare the melt between MITgcm and LADDIE. This is what I meant in "General comments"

when I meant that it is difficult to follow the different levels of evaluation. This paragraph is mixing too much. Can the authors restructure and clarify?

Agreed, we hope this confusion is resolved by the restructuring of the Results section

L416: About "the western side of the topographic channel": When talking North-East-South-West in Antarctica, it often gets confusing. Can the authors add labels for degrees East/West on the maps? Or they could add labels in the figure showing the "eastern flank" and "the western flank". That might make it easier to follow where to find the western side.

Good point, we will modify the figure to aid the orientation

Figure 6: I suggest annotating "atmosphere", "ice", "ocean" directly in at least one panel to make the figure better readable intuitively.

Good idea, we will implement this

L425: Would it make sense to add the results from MITgcm in Fig. 6 for a direct comparison and following better this sentence?

We have considered this suggestion, but as this cross-section would only encompass six grid cells from MITgcm, we consider the conclusion obvious from the plan-view map. We therefore will not implement this suggestion.

L426-444: These two paragraphs are a description and interpretation of features only resolved in LADDIE. I find these paragraphs difficult to interpret. LADDIE is used to make conclusions on processes, while it is being evaluated at the same time. Can we use it to interpret physical processes if we are not finished with the evaluation?

As mentioned above, we will follow this recommendation and separate the evaluation from the interpretation of new features.

L464-469: Again, what I find difficult here is the mixing between evaluation and interpretation, making it difficult to follow. I suggest to better separate them, for example: (1) evaluation, (2) what do we learn from LADDIE.

Agreed, see previous replies

L485: There is one "freezing" too much

Well spotted, will correct

L491-492: Is this a suggestion or have the authors looked at this into detail? If the latter, could the authors add one sentence how they came to this conclusion?

We will explain this by adding the following sentence: "As shown by Hausmann et al, the tides enhance freezing rates in this central region; a spatially heterogeneous tidal forcing could therefore reduce the discrepancy between LADDIE and observations."

L496: I suggest that this lack of spatial pattern comes, at least partially, from the use of 1D profiles as input, which the authors say in the following sentence. However, earlier, the authors suggested that using a 1D profile is acceptable compared to 3D fields. I think this shows that this assumption holds on small ice shelves, like Crosson-Dotson, but that, on large ice shelves, 1D profiles introduce more uncertainty than 3D fields.

This is a good point and we will acknowledge this.

L449: I do not completely agree with the conclusion that the qualitative large-scale pattern can be reproduced. It is good near the grounding line, yes, but the missing melting at the front, in the West, and at the southern tip of Berkner Island. I would suggest to acknowledge more that limitations remain.

Agreed, and will do

L515: Is it the goal to retune the parameter often? Again, this comes back to: Are the authors presenting a model to be used out of a box or is this more a proof of concept?

Again, we will include a discussion on future applications and the need for retuning

L557-564: This paragraph is confusing. I suggest reformulating or restructuring.

Agreed, we will reformulate this paragraph

L573: "more realistic forcing fields" => I again do not understand why realistic forcing fields were not used directly, especially if the authors suggest that it would have given better results.

See our previous replies

Discussion: If the authors do not plan to change their tuning evaluation setup, I suggest a clear and robust discussion paragraph about the influence of tuning and evaluating on the same ice shelf on the conclusions of the study.

We will indeed include a paragraph on the robustness of our tuning and evaluation method in the Methods section, where the tuning procedure is summarised. In addition, we will include simulations of Pine Island basal melt as additional evaluation of a warm ice shelf.

L581: I suggest replacing "idealised" with "existing"

We wish to emphasise that LADDIE is not a parameterisation, but a (simple) numerical model, and will hence retain the current formulation

L583: The list of citations is not very exhaustive here. Either add "e.g." in front or also include Lazeroms et al. 2018 and 2019, Favier et al. 2019, Beckmann and Goosse, 2002.

Although we already include 'e.g.,', we will add more citations for completeness

L591: "compared to basal melt parameterisations" => hmm, ok. But the authors have not compared the LADDIE results to "classic" parameterisations before in the

manuscript. Maybe add one or more citation where to find the patterns. Burgard et al. 2022 would be a possibility.

Agreed, we will add this citation

L599: Maybe because LADDIE was tuned towards reproducing Crosson-Dotson?

See previous replies regarding the comments on tuning

L637-645: Yes, the conclusions sound plausible but the paragraph is confusing. I suggest reformulating. In the end of this paragraph, I am left confused again about what was the precise goal of this study. I suggest the authors re-affirm in the beginning of the paper if the goal is a quantitative and/or qualitative evaluation. Due to the different levels of comparison, I was a bit lost.

Agreed, we will more clearly formulate the paper's aim and will rephrase this paragraph to make it clearer

Discussion: It would also help to add a paragraph about the planned application of LADDIE in the future. Would it be used for offline use or within an ocean model directly?

We will include this additional paragraph. Yet of course, it is not up to us to decide how others may use this model

L670-671: I agree. And would it not be also more consistent?

See previous replies on this aspect

L674: Can the authors do a similar comparison for FRISP? This would show how much of an influence the 1D vs 3D field has on a larger ice shelf.

Unfortunately, we were again unable to obtain these data from the authors. As we consider this of lower added value, we have not invested more effort into hunting these data down.

L682: "simulates" => "simulations"

Thanks, will correct

L732: Using the same parameters for both might be optimistic. Can the authors show similar figures for FRIS, just to compare, or does this involve tremendous effort?

This would involve the construction of new tuning targets specific for FRIS and a considerable number of simulations. As we show in this study, the same parameter settings (except for forcing parameters such as tidal velocity) produce good results for both Crosson-Dotson and Filchner-Ronne, which is a result in itself. We will therefore not pursue an additional tuning procedure for FRIS itself. However we do feel that a circum-Antarctic application of LADDIE, with associated tuning exercise, would be valuable future work, and we will mention this in the revised paper.

---

## Author Comment (AC3)

This paper introduces LADDIE, a 2D model that implements the depth-averaged navier stokes equations for ocean physics over a mixed layer thickness. The equations have been implemented before (e.g. Holland and Feltham 2006), but there are new modifications (for instance avoidig a hard constraint on minimum thickness), and moreover it is written in the form of an open source python code intended for wide use (though I have not tried it, and am not clear on how easy it is to port to another domain!) The model results are carefully compared against available observations for select ice shelves with high quality modelling and satellite observations of ice ocean interactions.

I have very few issues with this paper. The model itself is a step forward, and the reasons for it being a step forward are explained thoroughly, legibly, and carefully within the introduction, results and methods sections. The paper does make a very good point that there is a limit to how useful 3D ocean models can be due to cost and resolution required -- but is very clear on what LADDIE is *not* able to do ie model deep cavity dynamics -- and i found the discussion of its limitations (and possible extensions) to be very thougtful. As a scientific paper i feel adequate attention is given to the extensive literature on modelling and observing ice ocean interactions. I would recommend publication following the address of a few minor comments, which i feel will be quite easy.

two general but still minor comments are on the Dotson/Crosson results section:

a) there have been other studies with 3D ocean models run at higher resolution e.g. 1km, I wonder why you did not want to compare to these?

We have chosen the comparison to these ~3km resolution simulations based on their availability and their relatively long simulation times, making it possible to assess a quasi-steady state. We will phrase this more explicitly in the methods section.

b) Given the emphasis on the channelised melt, I think it is worth mentioning that a recent coupled modelling study (Goldberg and Holland 2022) saw this channel melt completely through within 50 years (in line with the extrapolation of Gourmelen 2017), and the ice-dynamic impact was minimal, somewhat downgrading its importance. The same is not true for the internal shear margin and grounding line of course!

Thank you for pointing out this relevant paper. We will refer to this to place the relevance of channelised melt into context.

line 34: "presumed stagnant" -- this is an assumption of LADDIE, not the physics of entrainment into the actual ML, which is how this reads.

Agreed, we will remove this classification

line 44: at-->over

Agreed, we will correct this

eqs 1-5: it would be nice to state whether these differ from the PDEs solved in Gladish et al 2012, and how if so. Also, this is for the author to decide, but an appendix showing how to do the integration that arises in the pressure terms (1st 2 terms on the LHS of (2) and (3)) would actually be quite helpful -- because im not sure i've ever seen clearly how these come about,

or how to do the layer integration and with which boundary conditions. For your consideration.

We appreciate this suggestion. However, these equations have been published in a considerable number of studies already. The primary addition of LADDIE to this research field is the numerical implementation of these equations, the development of an open-source model, and the application to realistic geometries. We therefore do not believe an additional derivation is required for this study. The pressure terms referred to are derived for a dense bottom boundary layer by Killworth and Edwards (1999), so we will add a reference to that derivation.

eqs 1, 8-10, and 14. Can you state that m_dot>0 indicates melt (if this is true). I don't think you do. (1) indicates it is, and i can reason this is consistent with (8) without referring to other papers. But i have not seen (14) before -- my simple understanding of it is that entrainment is enabled by positive TKE production, and (where there is freezing) by negative buoyancy flux. All seems consistent but it would be nice to be sure.

This is indeed true, and we will state it explicitly.

L192: i don't understand what a weighted average between free and no slip is. are you solving the model twice at each time step with different boundary conditions? would showing an equation help?

We have adopted this formulation of partial slip from the NEMO numerics and will expand its explanation.

L213: "one can interpret"... i think this is only true if the 3D model is isopycnal.

Agreed, we will mention this explicitly

L215: i like this rather than a hard constraint.

This is good to hear

L224-5: "to ensure continuity" -- by you, or the satellite analysts?

By the satellite analysts. We will rephrase this to clarify

L339: just to point out that these values for Ah are not huge but bigger, for instance, than that suggested by the MISOMIP protocol. What happens when you have Ah=5, do things change then? or is LADDIE unstable?

We agree that these values are still reasonable. Indeed, lower values of Ah lead to numerical instability which can either be resolved by a shorter time step or a smoother topography. Hence, LADDIE can perform the ISOMIP experiments with Ah=6 and Kh = 1. We will clarify this.

L400: "near-zero due to the lack of simulated barotropic flow" -- you don't show any evidence of this, or of it being the cause of low melt rates. My recollection is that the column here is quite a bit bigger than at the Smith and Pope grounding line. On the other hand the

Naughten model has pretty coarse vertical resolution at this depth and so the resolution of near-ice variation is particularly poor.. could this be a another potential reason?

Yes, we agree that we have not assessed this in detail, and the vertical resolution can certainly be a dominant factor. We will nuance this discussion and mention MITgcm's vertical resolution in this region as a possible explanation for the low melt rates.

Figure 6: could you show profiles from the 3D model as well?

Based on comments from reviewer 1, we have moved the discussion of 3D forcing to the appendix to avoid confusion. We therefore minimise the emphasis on this forcing and will not include these profiles.

line 444: what do you mean by the remote sensing not showing conclusive evidence? of channelised melt? or of specific features mentioned above? Im also not sure what you highlight in 3a -- there is very little detail here.

This notation referred to the separate meltwater pathways. We will rephrase this to avoid confusion.

L 458: it is possible that channelisation can lead to enhanced stresses and damage, but a reference would be nice here.

Agreed, we will add a reference

L495 -- where does this warm water come from? surface-warmed or other?

Indeed it is most likely that this is a surface-warmed water mass. We will mention this explicitly.

L534: propose-->suggest

Agreed and will implement

L608: a good point about subgl outflow. Is it not in fact trivial to add this?

Yes, this is indeed trivial to include if data is available. We will mention this

L620-623 -- a really good point about thin columns. Should note though this assumes detailed knowledge of bathymetry, which i think can only be this good if there is drilling, no?

Yes, we agree. We will clarify that this requires detailed knowledge of bathymetry

---

## Author Response (AR2)

Dear Editor,

Below, we copy the last comments from both reviewers and provide a point-by-point reply in blue.

**Reviewer #1: Clara Burgard**

I thank the authors for putting so much work into the revision of the manuscript! The new structure makes the reading more fluent and less confusing, and the main message is clearer. I also find the idea to put all the technical tuning information in the Appendix good. As the way the main messages are now formulated and there is an evaluation on Pine Island ice shelf, I find it more acceptable to both tune and evaluate on Dotson-Crosson ice shelf.

Dear reviewer,

We are happy that you are pleased with the revision of the manuscript. We agree that the current structure is a significant improvement upon the previous version.

I have two last minor comments.
(1) In L322, the authors refer to ESMs that do not resolve cavities. This information is important and I recommend mentioning it in the introduction, somewhere around L80.

This is a good suggestion, and we have mentioned this in the Introduction

(2) In L11, it is not completely clear what is meant by "without the need to retune parameters". Did the authors mean "without the need to retune parameters for each ice shelf individually"? I suggest clarifying.

Agreed, and we have specified this in the abstract

Thank you and looking forward to the opportunities that LADDIE opens, in particular for ice-sheet modelling!

**Anonymous reviewer #2**

I made only minor suggestions to the manuscript, and they have all been addressed.

however, one issue i have noticed with the code availability: the link
https://github.com/erwinlambert/laddie-description
is faulty.

Dear reviewer,

Thank you for your reviews. Unfortunately, we do not understand the problem with this link. We can open it properly and have asked a colleague to try to open the link, which he could. We therefore consider the link to be proper and have not modified it.